# SynCamMaster: Synchronizing Multi-Camera Video Generation from Diverse Viewpoints

**Jianhong Bai**[1*], **Menghan Xia**[2†], **Xintao Wang**[2], **Ziyang Yuan**[3], **Zuozhu Liu**[1], **Haoji Hu**[1†],
**Pengfei Wan**[2], **Di Zhang**[2]
[1]Zhejiang University, [2]Kuaishou Technology, [3]Tsinghua University

Project webpage: https://jianhongbai.github.io/SynCamMaster/

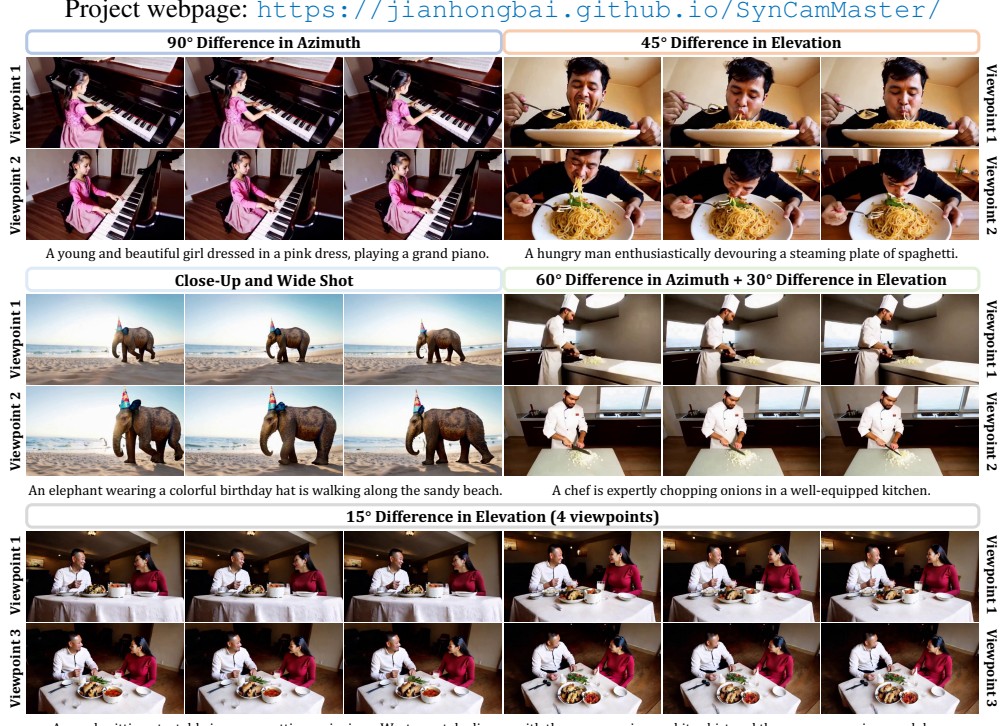

Figure 1: **Examples synthesized by SynCamMaster.** SynCamMaster generates multiple videos of the same dynamic scene from diverse viewpoints. Videos results are on our project page.

## Abstract

Recent advancements in video diffusion models have shown exceptional abilities in simulating real-world dynamics and maintaining 3D consistency. This progress inspires us to investigate the potential of these models to ensure dynamic consistency across various viewpoints, a highly desirable feature for applications such as virtual filming. Unlike existing methods focused on multi-view generation of single objects for 4D reconstruction, our interest lies in generating open-world videos from arbitrary viewpoints, incorporating 6 DoF camera poses. To achieve this, we propose a plug-and-play module that enhances a pre-trained text-to-video model for multi-camera video generation, ensuring consistent content across different viewpoints. Specifically, we introduce a multi-view synchronization module to maintain appearance and geometry consistency across these viewpoints. Given the scarcity of high-quality training data, we design a hybrid training scheme that leverages multi-camera images and monocular videos to supplement Unreal Engine-rendered multi-camera videos. Furthermore, our method enables intriguing extensions, such as re-rendering a video from novel viewpoints. We also release a multi-view synchronized video dataset, named SynCamVideo-Dataset. Our code is available at https://github.com/KwaiVGI/SynCamMaster.

---

* Work done during an internship at KwaiVGI, Kuaishou Technology. † Corresponding authors.

# 1 INTRODUCTION

As one of the most prominent generative models today, diffusion models have significantly advanced video generation technology. From the initial UNet-based explorations (Wang et al., 2023a; Blattmann et al., 2023; Chen et al., 2024; Wang et al., 2023b; Zhang et al., 2023a), to recent transformer-based scaling laws (Ma et al., 2024a; Yang et al., 2024b), state-of-the-art video diffusion models (Sora, 2024; Gen-3, 2024; Kling, 2024) are capable of producing dynamic simulations and 3D consistent videos that align with textual descriptions and adhere to real-world physical laws. This progress has sparked visions of future video creation paradigms, where controllable generation technology is poised to play an indispensable role in this evolution. Current explorations in controllable generation include structure control (Zhang et al., 2023b; Guo et al., 2023a), ID control (Jiang et al., 2024; Ma et al., 2024b), style control (Yang et al., 2023; Liu et al., 2023a), and single-camera control (Wang et al., 2024; He et al., 2024). However, multi-camera synchronized video generation, crucial for achieving combinational shots in virtual filming, remains under-explored.

Previous efforts in multi-camera generation have primarily focused on 4D object generation (Xie et al., 2024; Li et al., 2024). While these methods have shown promising results, they are limited to generating multi-view videos from fixed positions, such as sampling at equal intervals along an orbit around an object. Additionally, they are restricted to single-object domains and do not support open-domain scene generation. Recently, a concurrent work, CVD (Kuang et al., 2024), has explored synthesizing videos with multiple camera trajectories starting from the same pose. However, this approach has been studied only in the context of narrow viewpoints due to limitations in dataset construction. In this study, we focus on open-domain multi-camera video generation from arbitrary viewpoints. This task presents new challenges due to the goal of accommoding open-domain scenarios and the large discrepancies in viewpoints. Specifically, we face two main challenges: (i) dynamically synchronizing across multiple viewpoints, which introduces the complexity of maintaining 4D consistency, and (ii) the scarcity of multi-camera videos with diverse poses.

To address these challenges, we leverage the generative capabilities of a pre-trained text-to-video model by introducing plug-and-play modules. Specifically, given the extrinsic parameters of the desired cameras, normalized by setting one camera as the global coordinate system, we first encode these parameters into the camera embedding space using a camera encoder. We then perform inter-view feature attention computation within a multi-view synchronization module, which is integrated into each Transformer block of the pre-trained Diffusion Transformer (DiT). Additionally, we collected a hybrid training dataset consisting of multi-view images, general single-view videos, and multi-view videos rendered by Unreal Engine (Sanders, 2016) (UE). While the manually prepared UE data suffers from domain-specific issues and limited quantity, publicly available general videos enhance generalization to open-domain scenarios, and multi-view images promote geometric and visual consistency between viewpoints. Accordingly, we developed a tailored hybrid-data training scheme to optimize performance across these aspects.

Extensive experiments show that SynCamMaster can generate consistent content from different viewpoints of the same scene, and achieves excellent inter-view synchronization. Ablation studies highlight the advantages of our key design choices. Furthermore, our method can be easily extended for novel view synthesis in videos by introducing a reference video to our multi-camera video generation model. Our contribution can be summarized as follows:

- To our knowledge, SynCamMaster pioneered multi-camera real-world video generation.
- We design an efficient approach to achieve view-synchronized video generation across arbitrary viewpoints and support open-domain text prompts.
- We propose a hybrid-data construction and training paradigm to overcome the scarcity of multi-camera videos and achieve robust generalization.
- We extend our approach to novel view video synthesis to re-render an input video from novel viewpoints. Extensive experiments show the proposed SynCamMaster outperforms baselines by a large margin.

# 2 RELATED WORKS

**Controllable Video Generation.** With the success of text-to-video generation models (Blattmann et al., 2023; Wang et al., 2023b; Chen et al., 2024; Menapace et al., 2024), more accurate guidance

accompanied with text is often required in real-world applications. Previous research has successfully incorporated various conditional signals in video generation models, such as motion trajectory (Yin et al., 2023; Fu et al., 2024), sketch and depth (Guo et al., 2023a; Xing et al., 2024).

Several works (Guo et al., 2023b; Yang et al., 2024a; Bahmani et al., 2024) also integrate camera pose control into video diffusion models. AnimateDiff (Guo et al., 2023b) introduces various motion LoRAs (Hu et al., 2021) to learn specific patterns of camera movements. MotionCtrl (Wang et al., 2024) decouples camera motion and object movement and trains control modules to independently control both kinds of motion. CameraCtrl (He et al., 2024) further improves the accuracy and generalizability of single-sequence camera control with the dedicatedly designed camera encoder. The following work CVD (Kuang et al., 2024), expands CameraCtrl to multi-sequence camera control with the proposed cross-video synchronization module. Different from previous works, we focus on controllable multi-view video generation, rather than camera trajectory control in time dimension.

**Multi-View Image Generation.**   Multi-view image generation (Shi et al., 2023; Liu et al., 2023c; Kant et al., 2024) is well-studied in the scenario of 3D synthesis. Given one image as input, (Liu et al., 2023b;c; Chan et al., 2023) achieves 3D reconstruction by multiview-consistent image generation. ViewDiff (Höllein et al., 2024) enabling more realistic multi-view synthesis with background via training on real-world images and the proposed 3D projection layer. Despite the promising performance, it exhibits limited generalization capabilities.

**Multi-View Video Generation.**   Current studies on multi-view video generation primarily focus on 4D-asset synthesis (Liang et al., 2024; Li et al., 2024; Xie et al., 2024). SV4D (Xie et al., 2024) leverages both 3D priors in multi-view image generation models and motion priors in video generation models to synthesize multi-view videos from a reference video. Vivid-ZOO (Li et al., 2024) introduces trainable LoRAs (Hu et al., 2021) to alleviate the domain misalignment issue between 3D objects and real-world videos. In general, these approaches are expertise in object-level 4D synthesis, but could not generalized to real-world video with arbitrary viewpoints. A concurrent work GCD (Van Hoorick et al., 2024), dives into novel-view video synthesis in real-world scenarios, which is accomplished by training a video diffusion model conditioned on the camera pose and input video. Our method deviates from them in synthesizing multi-view videos with a single text prompt and desired viewpoints, while GCD generates monocular videos with an input video as the reference.

## 3   METHOD

Our goal is to achieve an open-domain multi-camera video generation model that can synthesize $n$ synchronized videos $\{\mathbf{V}^1, \ldots, \mathbf{V}^n\} \in \mathbb{R}^{n \times f \times c \times h \times w}$ with $f$ frames following the text prompt $P_t$ and $n$ specified viewpoints $\{\text{cam}^1, \ldots, \text{cam}^n\}$. The viewpoint is represented as the extrinsic parameters of the camera, i.e., $\text{cam}_i := [\mathbf{R}, \mathbf{t}] \in \mathbb{R}^{3 \times 4}$, where $\mathbf{R} \in \mathbb{R}^{3 \times 3}$ refers to the rotation matrix and $\mathbf{t} \in \mathbb{R}^{1 \times 3}$ is the translation vector. For simplification, we assume that the viewpoints remain constant across frames. To realize this, we propose to utilize the capability of pre-trained video diffusion models (Wang et al., 2023b; Chen et al., 2024) in 3D-consistent dynamic content synthesis and introduce a plug-and-play multi-view synchronization module to modulate the inter-view geometric and visual coherence. The overview of the model is depicted in Figure 2.

### 3.1   PRELIMINARY: TEXT-TO-VIDEO BASE MODEL

Our study is conducted over an internal pre-trained text-to-video foundation model. It is a latent video diffusion model, consisting of a 3D Variational Auto-Encoder (VAE) (Kingma & Welling, 2014) and a Transformer-based diffusion model (DiT) (Peebles & Xie, 2023). Typically, each Transformer block is instantiated as a sequence of spatial attention, 3D (spatial-temporal) attention, and cross-attention modules. The generative model adopts Rectified Flow framework (Esser et al., 2024) for the noise schedule and denoising process. The forward process is defined as straight paths between data distribution and a standard normal distribution, i.e.

$$z_t = (1 - t)z_0 + t\epsilon, \tag{1}$$

where $\epsilon \in \mathcal{N}(0, I)$ and $t$ denotes the iterative timestep. To solve the denoising processing, we define a mapping between samples $z_1$ from a noise distribution $p_1$ to samples $z_0$ from a data distribution

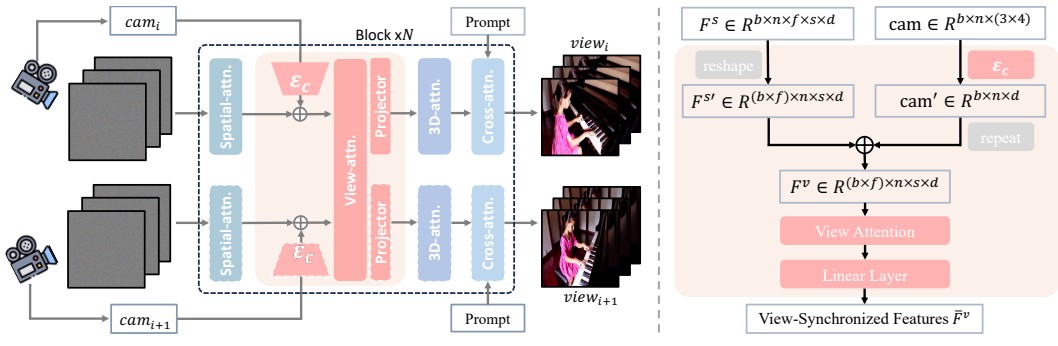

(a) Overview of SynCamMaster       (b) Multi-View Synchronization Module

Figure 2: **Overview of SynCamMaster.** Based on a pre-trained text-to-video model, two components are newly introduced: the camera encoder projects the normalized camera extrinsic parameters into embedding space; the multi-view synchronization module, as plugged in each Transformer block, modulates inter-view features under the guidance of inter-camera relationship. Only new components are trainable, while the pre-trained text-to-video model remains frozen.

$p_0$ in terms of an ordinary differential equation (ODE), namely:

$$dz_t = v_\Theta(z_t, t)dt, \tag{2}$$

where the velocity $v$ is parameterized by the weights $\Theta$ of a neural network. For training, we regress a vector field $u_t$ that generates a probability path between $p_0$ and $p_1$ via Conditional Flow Matching (Lipman et al., 2023):

$$\mathcal{L}_{LCM} = \mathbb{E}_{t,p_t(z,\epsilon),p(\epsilon)}||v_\Theta(z_t, t) - u_t(z_0|\epsilon)||_2^2, \tag{3}$$

where $u_t(z, \epsilon) := \psi'_t(\psi_t^{-1}(z|\epsilon)|\epsilon)$ with $\psi(\cdot|\epsilon)$ denotes the function of Eq. 1. For inference, we employ Euler discretization for Eq.2 and perform discretization over the timestep interval at $[0, 1]$, starting at $t = 1$. We then processed with iterative sampling with:

$$z_t = z_{t-1} + v_\Theta(z_{t-1}, t) * \Delta t. \tag{4}$$

## 3.2 MULTI-VIEW SYNCHRONIZATION MODULE

To achieve multi-view video synthesis, we train the multi-view synchronization (MVS) modules on top of the T2V generation model and leave the base model frozen. Note that, the operations below are conducted per frame across viewpoints, therefore we omit frame index $t$ for simplicity. The MVS module takes spatial features $\mathbf{F}^s = \{\mathbf{F}_1^s, \dots, \mathbf{F}_n^s\} \in \mathbb{R}^{n \times f \times s \times d}$ (where token sequence size $s = h * w$) and camera extrinsic parameters $\text{cam} = \{\text{cam}^1, \dots, \text{cam}^n\} \in \mathbb{R}^{n \times 12}$ of $n$ videos as input, and output view-consistent features $\overline{\mathbf{F}}^v = \{\overline{\mathbf{F}}_1^v, \dots, \overline{\mathbf{F}}_n^v\} \in \mathbb{R}^{n \times f \times s \times d}$ to the subsequent layers in the base T2V model.

Specifically, at each block in the video diffusion model, the 12-dimensional extrinsic parameters of the $i$-th camera are first embedded with a camera encoder $\mathcal{E}_c$ to have the same dimension as the spatial features, and then element-wise added to the corresponding spatial features. Then, we propose to leverage a cross-view self-attention layer for multi-view synchronization. Unlike layers in the T2V base model which operate within one single view feature, the additional cross-view attention layer takes multi-view features of the same frame as input, enabling cross-view feature aggregation. Finally, the aggregated features are then projected back to the spatial feature domain with a linear layer and residual connections. In summary, the multi-view synchronization (MVS) module is formulated as:

$$\mathbf{F}_i^v = \mathbf{F}_i^s + \mathcal{E}_c(\text{cam}^i), \tag{5}$$

$$\overline{\mathbf{F}}_i^v = \mathbf{F}_i^v + \texttt{projector}(\texttt{attn\_view}(\mathbf{F}_1^v, \dots, \mathbf{F}_n^v)[i]), \tag{6}$$

where we instantiate a fully connected layer with an input dimension of 12 and an output dimension of $d$ as camera encoder $\mathcal{E}_c$ in each block. Note that we insert the MVS module introduced above into each basic block of the DiT model to realize fine-grained control.

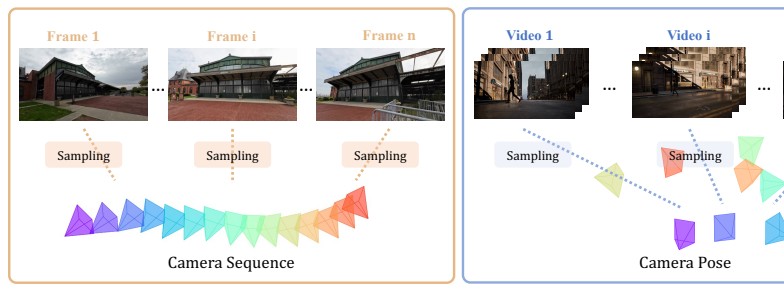

**(a). Construction of Multi-view Image Data**   **(b). Rendered Multi-view Video Data**   **(c). General Videos**

Figure 3: **Data collection process.** (a) Illustration of extracting multi-view image data from videos with camera movements, images are from DL3DV-10K (Ling et al., 2024); (b) Example of the rendered multi-view videos from diverse viewpoints; (c) Utilizing general video data as regularization.

### 3.3 DATA COLLECTION

The scarcity of multi-view video data is one of the main challenges hindering the training of multi-view video generation models. Existing multi-view video data primarily consists of **1)** videos rendered from 4D assets from various views (Liang et al., 2024; Zhang et al., 2024), and **2)** human-centric motion capture datasets (Ionescu et al., 2013; Joo et al., 2015; Shahroudy et al., 2016; Lin et al., 2022). However, these datasets do not adequately meet the needs of the task in this paper. For videos rendered from 4D assets (Liang et al., 2024), there is a significant domain gap between them and real-world videos, which can lead to severe degradation in video quality. For motion capture datasets, they mainly captured the videos from several fixed viewpoints (e.g., 31 viewpoints distributed over a hemispherical surface in Panoptic studio (Joo et al., 2015) and 4 in Human3.6M (Ionescu et al., 2013)), which hinders the model's generalization across arbitrary viewpoints.

To this end, we propose a three-step solution, as illustrated in Fig. 3. Firstly, we leverage single-camera videos as multi-view image data, and transfer the knowledge of geometry correspondence between different viewpoints to video generation. Specifically, RealEstate-10K (Zhou et al., 2018) and DL3DV-10K (Ling et al., 2024) contain videos with camera movements and their corresponding camera parameters across frames. We propose to sample $n$ video frames from the video as available multi-view image data. Secondly, we manually render a small number (500 scenes, 36 cameras each) of videos using the UE engine, featuring 3D assets such as humans and animals moving within urban environments. We enhance the model's generalization ability across arbitrary viewpoints by randomly placing camera positions. Lastly, we also incorporate high-quality general video data (without the corresponding camera information) as regularization during training. The construction process of our rendered multi-view video data is as follows.

**Construction of the Multi-View Video Data**
Firstly, we collect 70 3D assets of humans and animals as the main subjects and select 500 different locations in 3D scenes as background. Secondly, we randomly sample 1-2 main subjects to place them in each location and let them move along several pre-defined trajectories. Thirdly, we set up 36 cameras at different locations in each scene and rendered 100 frames synchronously. As a result, the multi-view video dataset is constructed of 500 sets of

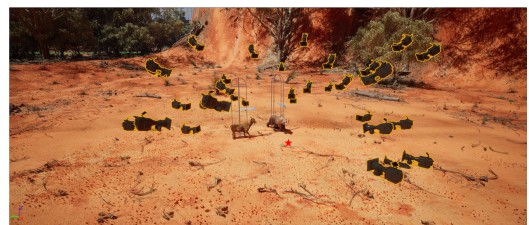

Figure 4: **Illustration of the rendering scene.**

synchronized videos with 36 cameras each. The cameras in each scene are placed on a hemispherical surface at a distance to the center of 3.5m - 9m. To ensure the rendered videos have minimal domain shift with real-world videos, we constraint the elevation of each camera between 0° - 45°, and the azimuth between 0° - 360°. To support SynCamMaster in synthesizing videos from arbitrary viewpoints, each camera is randomly sampled within the constraints, rather than using the same set of camera positions across scenes. Fig. 4 shows an example of one scene, where the red star indicates the center point of the scene (slightly above the ground), and the videos are rendered from the synchronized cameras to capture the movements of the main subjects (a goat and a bear in the

case). In the paper, we refer to this batch of rendered multi-view data as 'M. V. Video'. A detailed description and analysis of datasets are in Appendix B.

### 3.4 TRAINING STRATEGY

**Progressive Training.** To effectively learn the geometry correspondence between different viewpoints, we found it's crucial to start with feeding the model views with relatively small angle differences, and progressively increase the differences during training. Simply performing random sampling from different cameras in the same scene will lead to significant performance degradation in viewpoint following capabilities when input viewpoints with large relative angles (Fig. 7). A detailed description of training details is in Appendix C.

**Joint Training with Multi-View Image Data.** To alleviate the lack of multi-cam video data, we construct multi-view image data by sampling from single-camera video data as introduced in 3.3. We leverage DL3DV-10K (Ling et al., 2024) as the auxiliary image data, which includes ∼10K videos featuring wide-angle camera movements in both indoor and outdoor scenes. Our findings indicate that joint training with multi-view image data significantly enhances the generalization capability of SynCamMaster. This improvement is largely attributed to the diversity and scale of DL3DV-10K compared to our multi-view video data (10K vs. 500). Furthermore, the viewpoint-following capability is agnostic to the type of data, whether image or video, and is transferable between them.

**Joint Training with Single-View Video Data.** To improve the visual quality of the synthesized videos, we propose to incorporate high-quality video data (without camera information) as regularization. Given a single-view video, we augment it into multi-view videos by copying it $v$ times and setting the camera parameters the same across views. In other words, single-view videos are considered as videos with $v$ overlapping views during training. Moreover, we observe a performance degradation when simply using videos with arbitrary camera movements, it can be caused by distribution misalignment since SynCamMaster aims to generate videos from a fixed viewpoint. To this end, we filter out static camera video data using the following three steps: First, we downsample the video to 8 fps and use SAM (Kirillov et al., 2023) to segment the first frame, obtaining 64 segmentation masks. Next, we select the center point of each segmented region as the anchor point and use the video point tracking method CoTracker (Karaev et al., 2023) to calculate the position coordinates of each anchor point in all frames. Finally, we determine whether the displacement of all points is below a certain threshold. As a result, we filtered out 12,000 static camera videos, which were added as a regularization term during training.

### 3.5 EXTENSION TO NOVEL VIEW VIDEO SYNTHESIS

Additionally, we extend SynCamMaster to the task of novel view video synthesis (Van Hoorick et al., 2024; Zhang et al., 2024), which aims to generate videos captured from different viewpoints based on a reference video. To achieve this, we introduce modifications to turn SynCamMaster into a video-to-multiview-video generator. During training, given the noised latent features $\{z_t^1, \ldots, z_t^n\} \in \mathbb{R}^{n \times f \times c \times h \times w}$ of multi-view videos at timestep $t$, we regard the first view video as reference and replace the noisy latent of the video with its original one with the probability $p = 90\%$, i.e., $z_t^1 = z_0^1$. To this end, videos from novel views ($i = 2, \cdots, n$) could effectively aggregate features from the reference view via the proposed multi-view synchronization module introduced in Section 3.2. At the inference stage, we first extract the latent feature of the input video with the pre-trained video encoder and then perform feature replacement at each timestep $t = T, \cdots, 0$. Meanwhile, we implement weighted classifier-free guidance on text condition $c_T$ and video condition $c_V$ similar to Instruct-Pix2pix (Brooks et al., 2023):

$$
\begin{aligned}
\hat{v_\Theta}(z_t, c_V, c_T) = \; & v_\Theta(z_t, \varnothing, \varnothing) \\
& + s_V \cdot (v_\Theta(z_t, c_V, \varnothing) - v_\Theta(z_t, \varnothing, \varnothing)) \\
& + s_T \cdot (v_\Theta(z_t, c_V, c_T) - v_\Theta(z_t, c_V, \varnothing)),
\end{aligned}
\tag{7}
$$

where $s_T$ and $s_V$ are the weighting scores of the text condition and video condition respectively, we set $s_T = 7.5$ and $s_V = 1.8$ in practice. To this end, the proposed SynCamMaster can effectively re-render a video consistent with the text prompt and camera poses, as exhibited in Fig. 8.

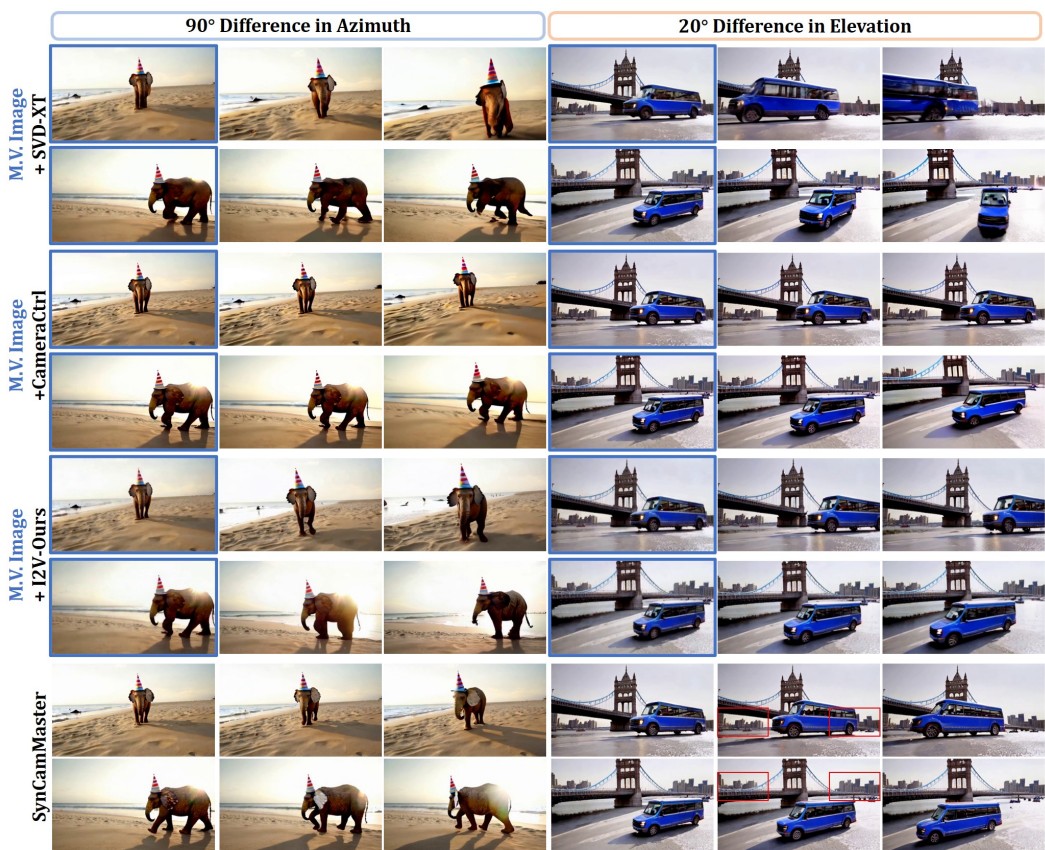

Figure 5: **Comparison with state-of-the-art methods.** The reference multi-view images of baseline methods (indicated in the blue box) are generated by SynCamMaster. It shows that SynCamMaster generates consistent content (e.g., the details in the red box) from different viewpoints of the same scene, and achieves excellent inter-view synchronization.

## 4 EXPERIMENTAL RESULTS

### 4.1 EXPERIMENT SETTINGS

**Implementation Details** We joint train our model on multi-view video data, multi-view image data, and single-view video data with the probability of 0.6, 0.2, and 0.2 respectively. We train the model of 50K steps at the resolution of 384x672 with a learning rate of 0.0001, batch size 32. The view-attention module is initialized with the weight of the temporal-attention module, and the camera encoder and the projector are zero-initialized.

**Evaluation Metrics** We mainly evaluate the proposed method in terms of cross-view synchronization and visual quality. In terms of cross-view synchronization, we utilize the state-of-the-art image matching method GIM (Shen et al., 2024) to calculate: 1) the number of matching pixels with confidence greater than the threshold, denoted as Mat. Pix., and 2) the average error between the rotation matrix and translation vector estimated by GIM of each frame and their ground truth, denoted as RotErr and TransErr respectively (He et al., 2024). Furthermore, we calculate the FVD-V score in SV4D (Xie et al., 2024), and the average CLIP similarity between multi-view frames at the same timestamp, denoted as CLIP-V (Kuang et al., 2024). For visual quality, we divide it into fidelity, coherence with text, and temporal consistency, and quantified them with Fréchet Image Distance (Heusel et al., 2017) (FID) and Fréchet Video Distance (Unterthiner et al., 2019) (FVD), CLIP-T, and CLIP-F respectively. CLIP-T refers to the average CLIP (Radford et al., 2021) simi-

Table 1: Quantitative comparison with state-of-the-art methods.

| Method | Visual Quality | | | | View Synchronization | | |
|---|---|---|---|---|---|---|---|
| | FID ↓ | FVD ↓ | CLIP-T ↑ | CLIP-F ↑ | Mat. Pix.(K) ↑ | FVD-V ↓ | CLIP-V ↑ |
| M.V. Image + SVD-XT | 137.3 | 1755 | - | 97.56 | 150.4 | 1742 | 89.14 |
| M.V. Image + CameraCtrl | 152.8 | 2203 | - | 98.32 | 172.9 | 1661 | 89.33 |
| M.V. Image + I2V-Ours | **113.1** | **1376** | **33.48** | 99.27 | 116.8 | 1930 | 90.01 |
| SynCamMaster | 116.7 | 1401 | 33.40 | **99.36** | **527.1** | **1470** | **93.71** |

Table 2: Quantitative ablation on the joint training strategy.

| Method | Visual Quality | | | | View Synchronization | | |
|---|---|---|---|---|---|---|---|
| | FID ↓ | FVD ↓ | CLIP-T ↑ | CLIP-F ↑ | Mat. Pix.(K) ↑ | FVD-V ↓ | CLIP-V ↑ |
| Multi-View Video | 149.9 | 1971 | 30.97 | 99.37 | 460.5 | 1668 | 89.68 |
| + Multi-View Image | 121.5 | 1655 | 33.02 | 99.36 | **533.0** | 1482 | 93.15 |
| + General Video | 122.4 | 1608 | 32.54 | **99.38** | 471.9 | 1514 | 90.12 |
| + Both | **116.7** | **1401** | **33.40** | 99.36 | 527.1 | **1470** | **93.71** |

larity of each frame and its corresponding text prompt, and CLIP-F is the average CLIP similarity of adjacent frames. We construct the evaluation set with 100 manually collected text prompts, and inference with 4 viewpoints each, resulting in 400 videos in total.

## 4.2 COMPARISON WITH STATE-OF-THE-ART METHODS

**Baselines** To the best of our knowledge, multi-view real-world video generation has not been explored by previous works. To this end, we establish baseline approaches by first extracting the first frame of each view generated by SynCamMaster, and then feeding them into 1) image-to-video (I2V) generation method, i.e., SVD-XT (Blattmann et al., 2023) 2) state-of-the-art single-video camera control approach CameraCtrl (He et al., 2024) based on SVD-XT. Since CameraCtrl would have non-optimal performance when conditions on static camera trajectory, we use a trajectory with limited movement as input. To ensure a fair comparison, we additionally train an I2V generation model based on the same T2V model used by SynCamMaster, the I2V model is fine-tuned with the approach similar to EMU Video (Girdhar et al., 2023) for 50K steps. During training, we extend and concatenate the latent feature of the first frame with the noised video latent along the channel dimension, and expand the dimension of the input convolutional layer with zero-initialized weights. We also replace the image latent with zeros at the probability of 0.1. At the inference stage, we implement weighted classifier-free guidance for the image and text condition (Brooks et al., 2023).

**Qualitative Results** We present synthesized examples of SynCamMaster in Fig. 1 (additional examples in Fig. 13-14 of Appendix E). Please visit our project page for more videos. SynCam-Master demonstrates the ability to: 1) generate consistent content from diverse viewpoints of the same scene; 2) achieve excellent inter-view synchronization; and 3) maintain the generation ability of base model across various text prompts.

We compare SynCamMaster with state-of-the-art methods in Fig. 5 (further comparisons in Fig.12 of Appendix E). Note that we use SynCamMaster to synthesize multi-view images (M.V. Images) as baseline methods' reference images (indicated in the blue box) since they cannot generate videos from various viewpoints. It's observed that the baseline methods failed to generate coherent videos across viewpoints. For example, the blue bus may stay in place in one shot, moving forward in another. While SynCamMaster can synthesize view-aligned videos that adhere to both the camera poses and text prompts.

**Quantitative Results** We quantitatively evaluate our method using various automatic metrics, the summarized results are in Tab. 1. For view synchronization, we calculate the clip similarity score and FVD between video frames of different viewpoints within one scene, denoted as CLIP-V and

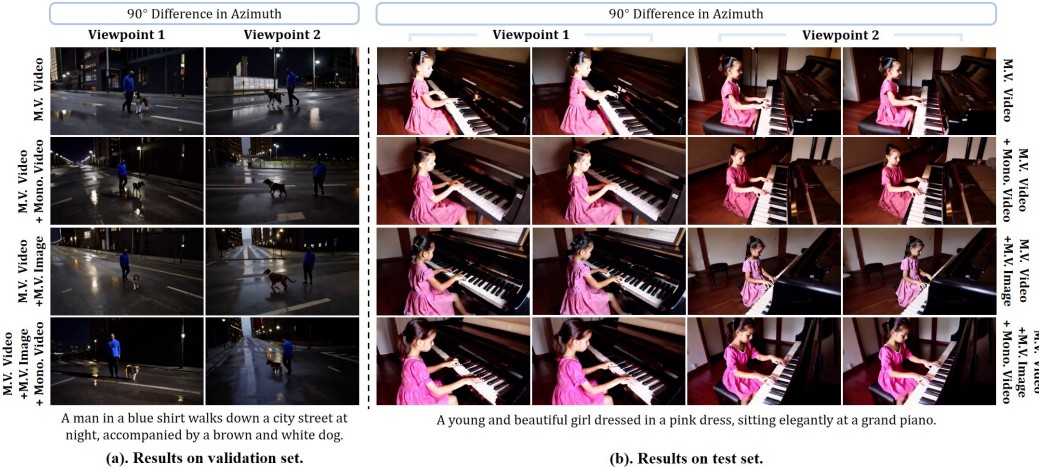

Figure 6: **Ablation on the joint training strategy.** The captions on both sides represent the composition of the training set, where "Mono. Video" refers to general monocular videos. It shows that training with the auxiliary multi-view image data and general video data significantly improves the generalization ability and fidelity of the synthesized videos.

Table 3: Results of novel view video synthesis.

| Setting | LPIPS ↓ | PSNR ↑ | SSIM ↑ |
|---|---|---|---|
| $s_V = 1.2, s_T = 5.0$ | 0.4899 | 16.29 | 0.4754 |
| $s_V = 1.2, s_T = 7.5$ | 0.4901 | **16.60** | 0.4783 |
| $s_V = 1.8, s_T = 7.5$ | **0.4761** | 16.47 | **0.4935** |
| $s_V = 2.5, s_T = 7.5$ | 0.5022 | 14.55 | 0.4667 |

Table 4: Accuracy of camera control.

| Method | RotErr ↓ | TransErr ↓ |
|---|---|---|
| M.V. Image + SVD-XT | 0.25 | 0.72 |
| M.V. Image + CameraCtrl | 0.16 | 0.67 |
| M.V. Image + I2V-Ours | 0.26 | 0.80 |
| SynCamMaster | **0.12** | **0.58** |

FVD-V. It's observed that the proposed SynCamMaster significantly outperforms baseline methods, which is aligned with qualitative results in Fig. 5. To further eliminate the effect of different base models, we also fine-tuned an I2V generation model on top of the same base model used by SynCam-Master, denoted as 'I2V-Ours'. As observed, SynCamMaster achieves comparable performance in terms of fidelity, text alignment, and frame consistency, while having better synchronization across views. Furthermore, we use 100 videos containing camera pose differences in azimuth or elevation to calculate the accuracy of camera control. The average error along frames is reported in Tab. 4, where SynCamMaster has superior performance compared to baselines. Note that we normalize the relative translation vector of two viewpoints to have a norm of 1.0 to align the scale between the estimated poses and ground truth (Xu et al., 2024).

## 4.3 MORE ANALYSIS AND ABLATION STUDIES

**The Effectiveness of Joint Training** As introduced in Section 3.4, we utilize multi-view images and general videos to alleviate the scarcity of available multi-view video data, and further improve the generalization ability and visual quality of SynCamMaster. To verify the effectiveness, we make qualitative comparisons in Fig. 6 and exhibit quantitative results in Tab. 2. In Fig. 6(a), we visualize the generation results of the proposed method on the validation set by conditioning on two cameras with a 90-degree difference in azimuth. It's observed that only training SynCamMaster on the rendered multi-view video dataset is sufficient to generate synchronized videos on the training domain, which demonstrates the proposed multi-view synchronization module could effectively generate view-consistent features. However, due to the domain gap between the rendered data and the diverse real-world videos, merely training on multi-view videos could result in poor performance when transferring the multi-view synthesis ability to general videos. As shown in Fig. 6(b), we observe a severe performance degradation in pose following and synchronization when inference with test set prompts. In this case, the girl's arms are at different positions across the two viewpoints, with the camera pose misalignment. The issue is significantly alleviated when we engage multi-view image data in training, as exhibited in the second line. Furthermore, we enhance the fidelity of SynCamMaster with diverse general video data.

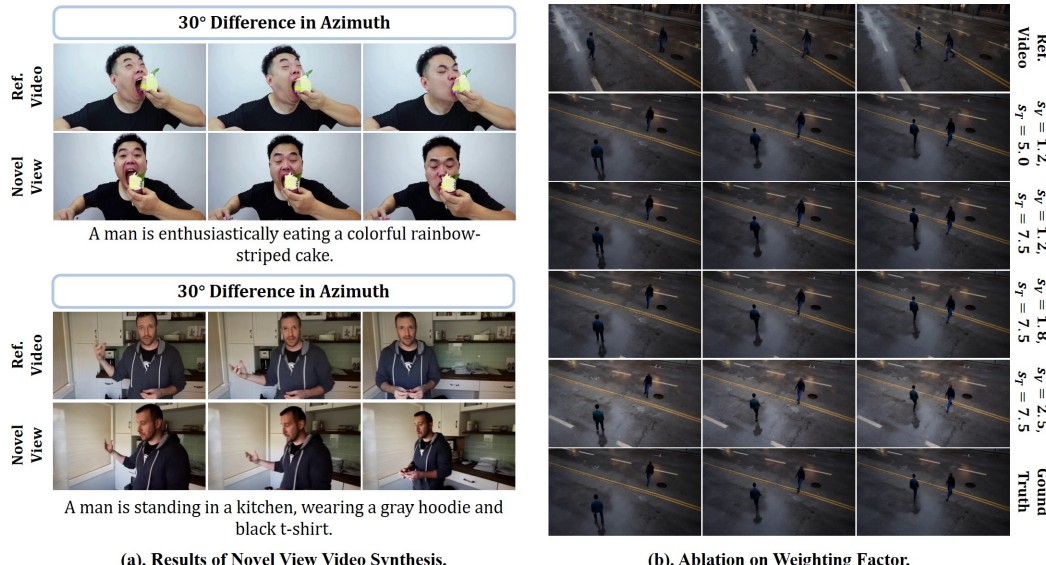

(a). Results of Novel View Video Synthesis.

(b). Ablation on Weighting Factor.

Figure 8: **Results of the extension on novel view video synthesis.**

**The Effectiveness of Progressive Training** View synchronized synthesis is particularly challenging when aiming to generate videos with large viewpoint differences since there are fewer matching features across views. To tackle this problem, we propose a progressive training strategy that gradually increases the relative angle between different viewpoints during training. In Fig. 7, we visualize the synthesized results of training w/ or w/o the proposed progressive training strategy. It's observed that while simply training with samples that have arbitrary relative angles can also generate view-consistent videos, they would fail in pose-following ability when conditioning on cameras with large relative angles.

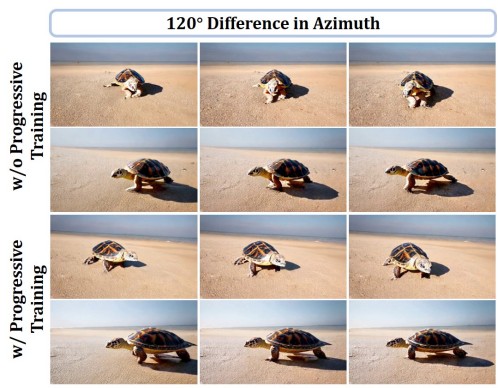

Figure 7: **Ablation on progressive training.**

**Ablation on Classifier-Free Guidance in Novel View Video Synthesis** In Section 3.2, We extend the proposed text-to-multiview video generation method to novel view video synthesis, i.e., video-to-multiview video generation. We present the synthesized videos in Fig. 8(b) on the validation set with different sets of weighting scores $s_V$ and $s_T$ in Eq. 7. As shown, SynCamMaster can re-render the input video at arbitrary viewpoints, while maintaining excellent view synchronization by setting the weighting scores in a certain range. Quantitatively, we calculate the LPIPS, PSNR, and SSIM scores on 100 randomly selected video pairs in Tab. 3, and we observe comparable results with a previous work GCD (Van Hoorick et al., 2024). We set $s_V = 1.8$ and $s_T = 7.5$ in practice.

## 5 CONCLUSION AND LIMITATIONS

In this paper, we propose SynCamMaster to generate synchronized real-world videos from arbitrary viewpoints. To achieve this, we leverage the pre-trained text-to-video generation model and design a multi-view synchronization module to maintain appearance and geometry consistency across different viewpoints. We also extend our method to novel view video synthesis to re-render an input video. There are nevertheless some limitations. Firstly, when generating videos with complex scenes, the content in the synthesized multi-view videos may exhibit inconsistencies in details. Secondly, since SynCamMaster is based on T2V models, it also inherits some of the shortcomings of the base model, such as inferior performance in hand generation. We exhibit the failure cases in Fig. 15.

## ACKNOWLEDGMENTS

We thank Jinwen Cao, Yisong Guo, Haowen Ji, Jichao Wang, and Yi Wang from Kuaishou Technology for their invaluable help in constructing the multi-view video dataset. We thank the authors of 4DGS (Wu et al., 2024) and Jiangnan Ye for their help in running 4D reconstruction.

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

# A    INTRODUCTION OF THE BASE TEXT-TO-VIDEO GENERATION MODEL

Figure 9: Overview of the base text-to-video generation model.

We use a transformer-based latent diffusion model (Peebles & Xie, 2023) as the base T2V generation model, as illustrated in Fig. 9. Initially, we employ a 3D-VAE to transform videos from the pixel space to a latent space, upon which we construct a transformer-based video diffusion model. Unlike previous models that rely on UNets or transformers, which typically incorporate an additional 1D temporal attention module for video generation, such spatially-temporally separated designs do not yield optimal results. We replace the 1D temporal attention with 3D self-attention, enabling the model to more effectively perceive and process spatiotemporal tokens, thereby achieving a high-quality and coherent video generation model. Specifically, before each attention or feed-forward network (FFN) module, we map the timestep to a scale, thereby applying RMSNorm to the spatiotemporal tokens.

# B    DATA CONSTRUCTION

## B.1    TRAINING DATASETS

Recall in Section 3.3, we train the proposed SynCamMaster with data from different sources. In this section, we provide a detailed description of the construction process of various datasets.

**Multi-View Video Data**    Firstly, we collect 70 3D assets of persons and animals as the main subjects and select 500 different locations in 3D scenes as background. Secondly, we randomly sample 1-2 main subjects to place them in each location and let them move along several pre-defined trajectories. Thirdly, we set up 36 cameras at different locations in each scene and rendered 100 frames synchronously. As a result, the multi-view video dataset is constructed of 500 sets of synchronized videos with 36 cameras each.

The cameras in each scene are placed on a hemispherical surface at a distance to the center of 3.5m-9m. To ensure the rendered videos have minimal domain shift with real-world videos, we constraint the elevation of each camera between $0°$ - $45°$, and the azimuth between $0°$ - $360°$. To support SynCamMaster in synthesizing videos from arbitrary viewpoints, each camera is randomly sampled within the constraints described above, rather than using the same set of camera positions across scenes. Fig. 4 shows an example of one scene, where the red star indicates the center point of the scene (slightly above the ground), and the videos are rendered from the synchronized cameras to capture the movements of the main subject.

**Multi-View Image Data** In this paper, we construct multi-view image data from DL3DV-10K (Ling et al., 2024). DL3DV-10K is composed of 10,510 videos from both indoor and outdoor scenes, with their corresponding camera poses estimated by structure from motion (SFM) methods. We utilize its 960P version and downsample the frames to the resolution of 384x672.

**General Video Data** We also utilize single-view video data to further improve the visual quality of the synthesized videos. We observe a performance degradation when simply using videos with arbitrary camera movements, it can be caused by distribution misalignment since SynCamMaster aims to generate videos from a fixed viewpoint. To this end, we filter out static camera video data using the following three steps: First, we downsample the video to 8 fps and use SAM (Kirillov et al., 2023) to segment the first frame, obtaining 64 segmentation masks. Next, we select the center point of each segmented region as the keypoint and use the video point tracking method CoTracker (Karaev et al., 2023) to calculate the position coordinates of each keypoint in all frames. Finally, we determine whether the displacement of all points is below a certain threshold. As a result, we filtered out 12,000 static camera videos, with an average duration of 9 seconds, which were added as a regularization term during training.

## B.2 DISCUSSION ON OTHER MULTI-VIEW DATASETS

**Multi-View Images** Multi-view images are also utilized in image novel view synthesis. Co3D (Reizenstein et al., 2021) and MVImgNet (Yu et al., 2023) are object-centric datasets including multi-view frames from multiple object classes. Despite their large scale, they do not meet the requirements of our task in two aspects. On the one hand, we aim to synthesize multi-view real-world videos, which have a domain gap between the object-centric frames. On the other, most of the backgrounds in these datasets do not have obvious features. For example, most objects are placed on solid-colored tables or roads, which makes it more challenging to learn geometry correspondence. We observe inferior performance in terms of camera pose following ability when integrating Co3D and MVImgNet into training. Furthermore, similar to DL3DV-10K, RealEstate-10K (Zhou et al., 2018) is also a commonly used dataset at scene level. Compared to RealEstate-10K, DL3DV-10K has more videos with rotating perspectives, which benefits SynCamMaster from synthesizing videos with large differences in amizuth.

**Multi-View Videos** Previous works have established frame-synchronized multi-view video data (Ionescu et al., 2013; Joo et al., 2015) for human pose estimation, action recognition, etc. Nevertheless, the cameras are fixed across different videos, which hinders SynCamMaster's ability to learn to generate videos from arbitrary viewpoints. Additionally, some recent works (Xie et al., 2024; Zhang et al., 2024) obtain multi-view video by filtering out 4D assets from Objaverse (Deitke et al., 2023) and rendering them with multiple cameras. While these data are helpful for 4D object generation, the significant domain gap is insufficient to support the generation of multi-view real-world videos.

## C IMPLEMENTATION DETAILS

**Sampling Strategy** To effectively learn the geometric correspondences between different viewpoints, we have carefully designed a sampling strategy during training. For multi-view image data, to ensure overlap in the field of view between selected frames, we limit the maximum distance between $v$ different frames to 100 frames during sampling. For multi-view image data, we first calculate the azimuth angle of each camera, and then ensure that the relative elevation angles between each pair of the selected $v$ video clips fall within the interval $[\theta_l, \theta_h]$. The values of $\theta_l$ and $\theta_h$ are determined based on the number of training steps. In practice, we use $\theta_l = 0, \theta_h = 60$ for 0-10K steps, $\theta_l = 30, \theta_h = 90$ for 10K-20K steps, and $\theta_l = 60, \theta_h = 120$ for steps greater than 20K. We precompute all possible combinations before training to enhance efficiency.

**Training Configurations** The proposed SynCamMaster is jointly trained on multi-view video, multi-view image, and general video data as introduced in Section 3.3. At the beginning of each step, we randomly select the type of data to be used for that step based on predefined probabilities (0.6, 0.2, 0.2 in practice). We use a batch size of 32 and a learning rate of 0.0001 to train the model for 50k steps at the resolution of 384x672.

**Construction of Evaluation Set**    To evaluate SynCamMaster's accuracy on camera control, we set up several groups of camera poses with variations in either azimuth or elevation angles. We used the image matcher method GIM (Shen et al., 2024) to obtain the estimated homography matrix, which was further decomposed to derive the relative extrinsic matrix. Then we computed the rotation error and translation error respectively by following CameraCtrl (He et al., 2024) and CamCo (Xu et al., 2024). Specifically, under the setting of generating four synchronized videos ($v = 4$), we configured adjacent cameras with azimuth angle differences of $\{10°, 20°, 30°\}$ or elevation angle differences of $\{10°, 15°\}$, resulting in a total of 5 different camera setups, with 20 videos per setup. The evaluation results are summarized in Tab. 4.

## D    MORE ANALYSIS AND ABLATION STUDIES

**The Choice of Camera Representation**    In this paper, we used the camera extrinsic matrix as the camera representation. The ray positional embedding is also widely used in 3D reconstruction (Gao et al., 2024) and camera control (He et al., 2024) techniques. We evaluate the impact of using different representations on camera accuracy, and present the qualitative and quantitative results in Fig. 10 and Tab. 5 respectively. As shown, there is no significant difference in the generated results between the two representations. We assume this is because the UE data (rendered multi-camera synchronized videos) has the same camera intrinsics, and in this case, both representations contain consistent information.

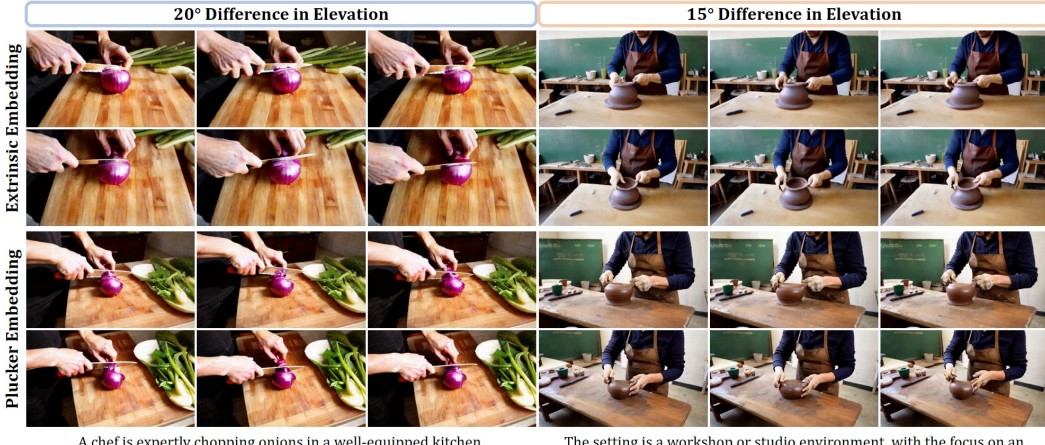

A chef is expertly chopping onions in a well-equipped kitchen.    The setting is a workshop or studio environment, with the focus on an individual engaged in pottery-making.

Figure 10: Comparasion on using different camera representations.

Table 5: Accuracy of camera control with different camera representations.

| Camera Representation | RotErr ↓ | TransErr ↓ |
|---|---|---|
| Plucker Embedding | **0.12** | 0.60 |
| Extrinsic Embedding | **0.12** | **0.58** |

**Ablation on Epipolar Attention**    To effectively learn the spatial geometry, previous studies (He et al., 2020; Kant et al., 2024) explicitly model 3D information by using epipolar-constrained attention layers. In Fig. 11 and Tab. 6, we compare the results w/ and w/o epipolar attention in view attention layers. Specifically, when implementing epipolar attention, for the token at spatial position $(x, y)$ in view $i$, it only aggregates features from all tokens within view $i$ and tokens on the epipolar line in other views. In contrast, each token attends to all tokens in all views in the full attention setting. We found that although epipolar attention has low rotation error, it can result in inconsistency with the text prompt semantics (shown in Fig. 11) compared to the full-attention design. On the other, we found that it is sufficient to learn the spatial correspondence in a data-driven manner without explicit 3D modeling, which is consistent with the findings of recent work (Jin et al., 2024).

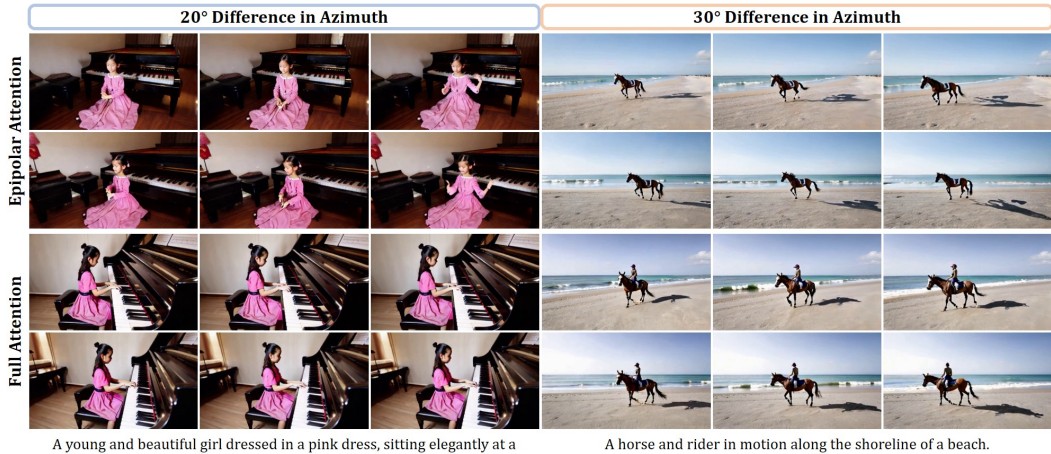

A young and beautiful girl dressed in a pink dress, sitting elegantly at a grand piano.

A horse and rider in motion along the shoreline of a beach.

Figure 11: Performance comparison of SynCamMaster with epipolar attention and full attention.

Table 6: Accuracy of camera control with epipolar attention and full attention.

| Camera Representation | RotErr ↓ | TransErr ↓ |
|---|---|---|
| Epipolar Attention | **0.10** | 0.59 |
| Full Attention | 0.12 | **0.58** |

**Discussion on the Data Mixture Strategy**  In the paper, we jointly train our model on multi-view video data, multi-view image data, and single-view video data with probabilities of 0.6, 0.2, and 0.2, respectively. We also explored the impact of different mixing ratios on the generated results. We found that a higher proportion of multi-view image data disrupts the temporal continuity of the videos. On the other hand, using too much single-view video data causes the model to favor synthesizing views with small relative angles, affecting the camera accuracy. Therefore, we sample multi-view image data and single-view video data with small probabilities.

# E   MORE RESULTS

## E.1   MORE COMPARISON WITH STATE-OF-THE-ART METHODS

In addition to using FID, FVD, and CLIP scores to evaluate the visual quality of the generated videos in Tab. 1, we also assessed the generation quality of our method and the baselines using the video generation evaluation method VBench (Huang et al., 2024) as supplementary. Specifically, we randomly sample 100 prompts from the 300 prompts provided by VBench in the categories of 'animal', 'human', and 'vehicles' for evaluation. We did not sample from categories such as 'plant' and 'scenery' because, given that we generate videos from fixed camera positions, static scenes would result in videos that are stationary over time. The evaluation results are presented in Tab. 7. Our method outperforms SVD-XT (Blattmann et al., 2023) and CameraCtrl (He et al., 2024) across all metric dimensions and demonstrates performance comparable to our trained I2V model.

Table 7: Quantitative comparison with baseline methods on VBench (Huang et al., 2024).

| Method | Subject Consistency | Background Consistency | Aesthetic Quality | Imaging Quality | Temporal Flickering | Motion Smoothness |
|---|---|---|---|---|---|---|
| M.V. Image + SVD-XT | 94.32 | 94.23 | 48.85 | 52.80 | 95.79 | 98.26 |
| M.V. Image + CameraCtrl | 95.91 | 96.40 | 50.44 | 52.85 | 96.79 | 98.73 |
| M.V. Image + I2V-Ours | 93.53 | 92.93 | 49.07 | **58.49** | 95.20 | 98.13 |
| SynCamMaster | **97.84** | **96.55** | **50.50** | 58.30 | **98.95** | **99.27** |

Please refer to Fig. 12 for qualitative comparison with the state-of-the-art methods.

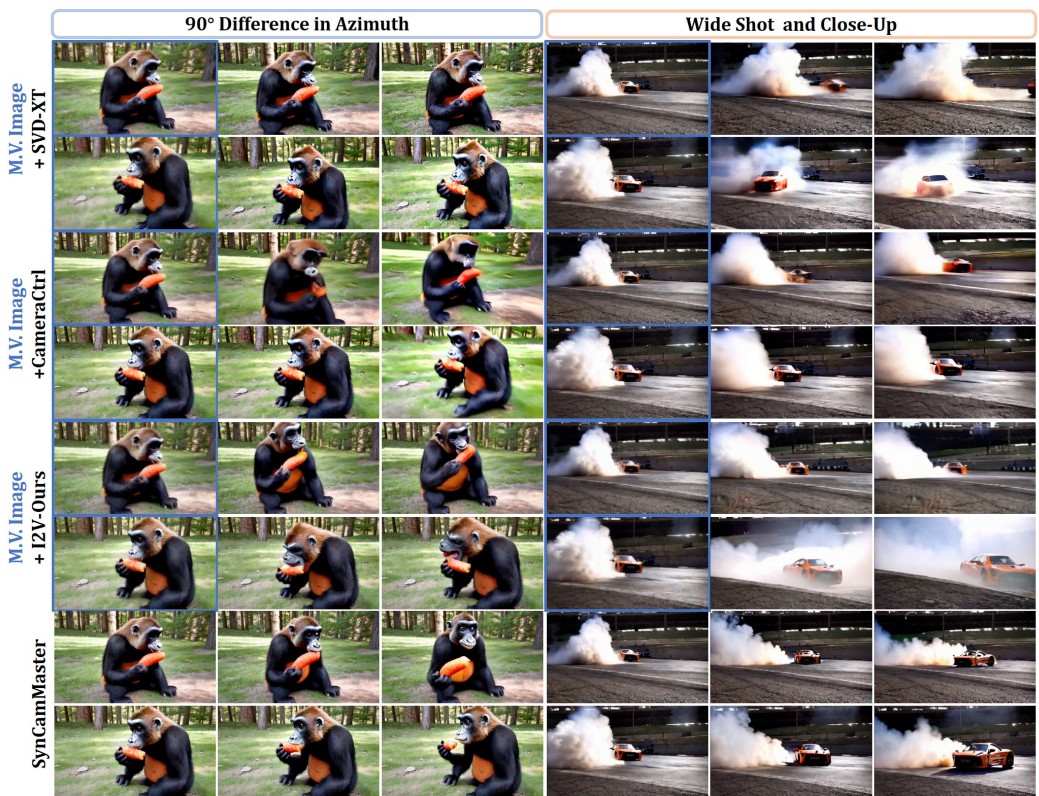

Figure 12: More comparison with state-of-the-art methods.

### E.2 MORE RESULTS OF SYNCAMMASTER

More synthesized results of SynCamMaster are presented in Fig. 13-14.

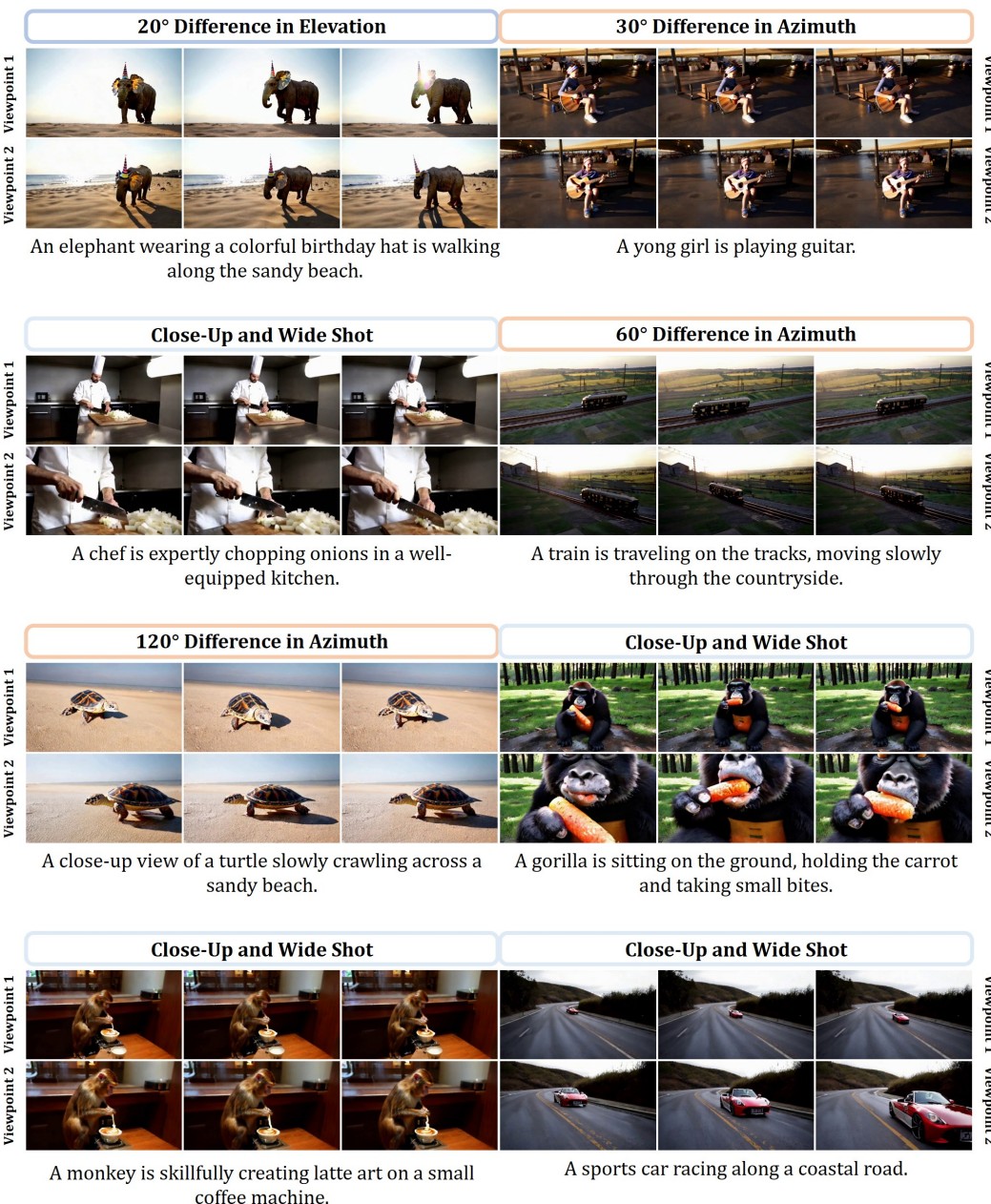

Figure 13: More synthesized results of SynCamMaster.

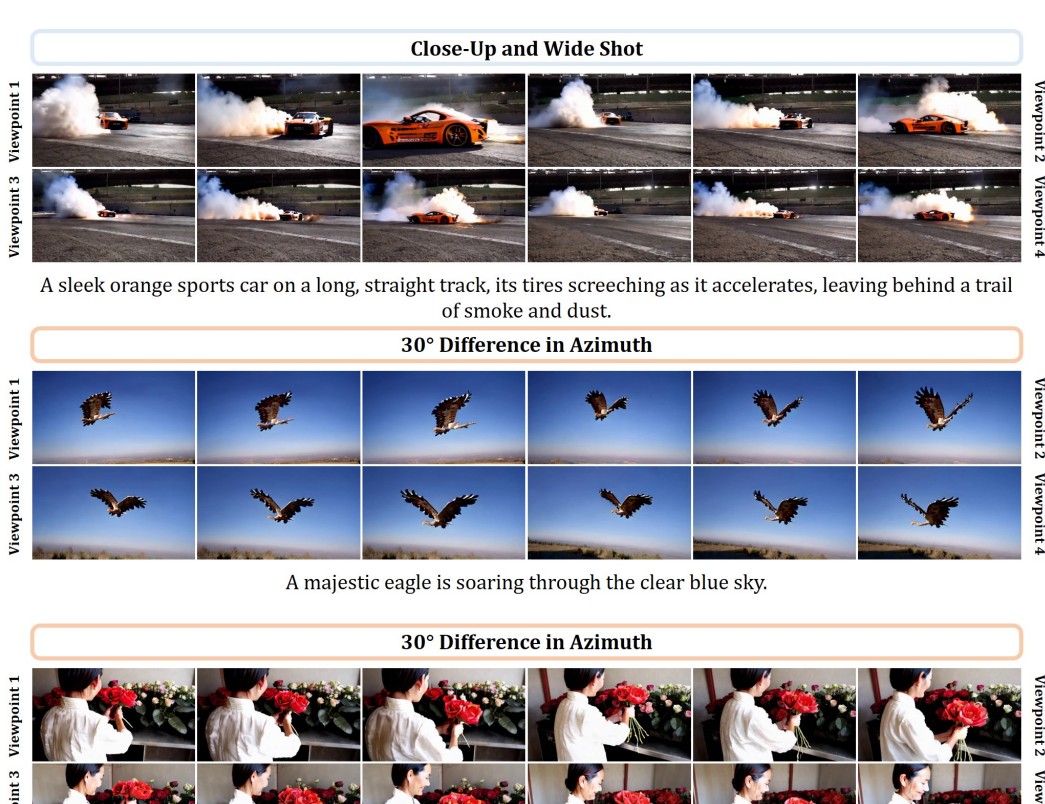

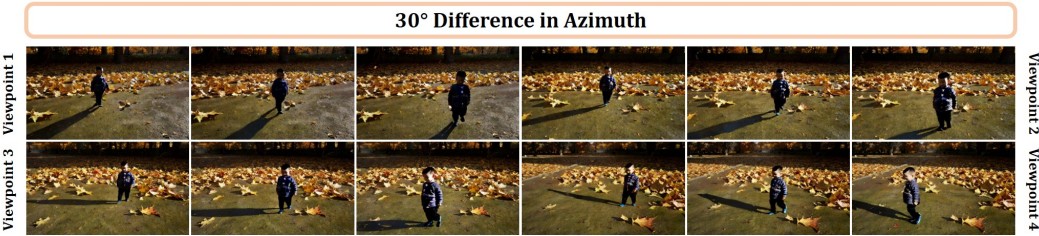

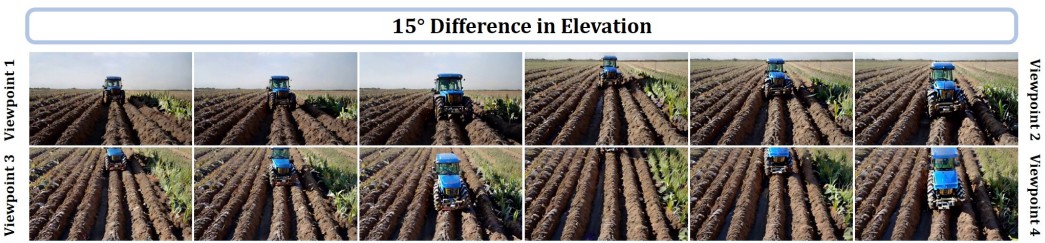

Figure 14: More synthesized results of SynCamMaster.

### E.3    FAILURE CASES VISUALIZATION

We present the failure cases in Fig. 15. Firstly, we observe that inconsistencies in details may occur when generating complex scenes, for example, in the first and second rows of Fig. 15, the bowls and plates on the table show content discrepancies between the two viewpoints. Secondly, since our model is built upon a text-to-video base model, we also inherit some of the base model's shortcomings. For instance, the generated hand movements of characters may exhibit inferior quality, as shown in the third and fourth rows.

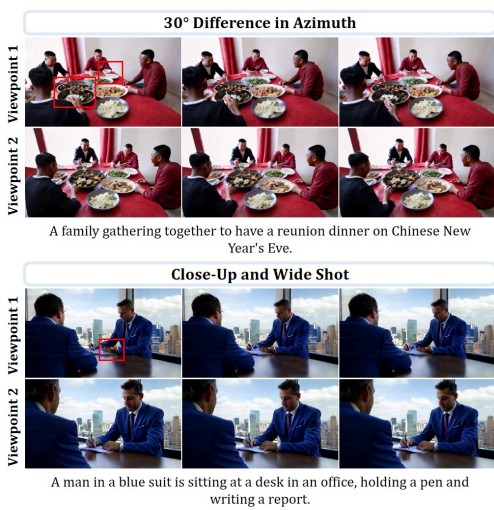

Figure 15: **Visualization of failure cases.**

