# OpenReview forum: "SynCamMaster: Synchronizing Multi-Camera Video Generation from Diverse Viewpoints"
_ICLR.cc/2025/Conference — ICLR 2025 Poster_

### Official Review · Reviewer_kLvx · 2024-11-04

**Soundness:** 3
**Presentation:** 3
**Contribution:** 3
**Rating:** 6
**Confidence:** 4

**Summary:**

This paper introduces a plug-and-play module for generating consistent videos from arbitrary viewpoints using 6 DoF camera poses. Unlike methods focused on 4D reconstruction of single objects, this approach enhances a pre-trained text-to-video model for synchronized multi-camera generation, ensuring appearance and geometry consistency across viewpoints. To overcome limited training data, this paper uses a progressive training scheme, leveraging multi-camera and monocular videos along with Unreal Engine-generated data.

**Strengths:**

1. The approach of generating synchronized multi-view videos is a strong point, as these videos offer applications beyond traditional 4D reconstruction, such as enhancing virtual filming. By maintaining dynamic consistency across views, this method broadens the potential use cases in various real-time and interactive settings.

2. The visual quality of the generated videos is impressive, particularly with respect to the wide camera baselines achieved. This indicates the model’s robustness in handling significant viewpoint shifts, preserving both appearance and geometric consistency across diverse perspectives, which is essential for applications requiring high-fidelity visual continuity.

3. Leveraging multi-view video data from game engines for fine-tuning is a smart approach, especially given the scarcity of high-quality multi-camera real-world datasets. The method demonstrates effective generalization to real-world data, showing that game engine data can be a viable supplement for improving model performance in open-world video generation tasks.

**Weaknesses:**

1. Although 4D video reconstruction is highlighted as a key application, the paper does not provide specific results on this. Including an evaluation of 4D reconstruction performance (like 4DGS) on the output synchronized videos would strengthen the contribution and showcase the practical utility of the generated multi-view videos in real-world 4D reconstruction tasks.

2. The use of camera conditioning is potentially limited, as it may not handle large out-of-distribution camera pose translations during inference (e.g., for training only 3.5m - 9m shifts). Testing the model on such out-of-distribution poses would provide valuable insight into its robustness and adaptability to extreme viewpoint variations, which are common in real-world applications.

3. Multi-view consistency could be better assessed. A qualitative evaluation, such as a cycle consistency check, could help—rendering videos from generated camera poses and then re-rendering from the original poses would allow comparison. The differences between these two renderings could serve as a metric for the model’s ability to maintain consistent content across varying views, giving a clearer measure of its multi-view coherence.

**Questions:**

1. Will the data preparation process and resulting dataset be made publicly available? The paper states that data was collected from a game engine, which could be valuable for advancing research in multi-view video generation. Public access to this data would enable reproducibility, facilitate benchmarking, and help other researchers develop and test models in a similar setting.

2. How were the data mixture probabilities (0.6, 0.2, 0.2) chosen, and what role do they play in the model’s performance? It would be helpful to understand the rationale behind this distribution and whether alternative ratios were explored. Insights into how these probabilities impact training dynamics and model generalization would also clarify the importance of the data mixture balance in achieving optimal results.

**Details Of Ethics Concerns:**

Human faces are shown in the paper. Not sure if they are generated or sampled from some datasets.

---

> ### Author Response · Authors · 2024-11-21
> **Response to Reviewer kLvx**
>
> We thank the reviewer for the insightful comments, and we address the concerns below:
>
> >Evaluate the generated videos by 4D reconstruction.
>
> Thanks for the advice! We add the 4D reconstruction results with 4DGS [1] to our project page (https://syncammaster.github.io/SynCamMaster/). Specifically, we condition SynCamMaster on four cameras with 10° difference in azimuth and the text prompt to synthesize multi-view videos. Then, we use the four view camera parameters and our generated videos as input to 4DGS [1], and follow their open-source code (https://github.com/hustvl/4DGaussians) to train the 4D Gaussian splatting. As exhibited, the rendered novel views are overall consistent with fewer artifacts.
> It's worth noting that it's quite challenging to reconstruct 4D with sparse views (4 views in the case) at low resolution (384x672 in the case), therefore the performance can be further improved by 1) utilizing a T2V model that is able to generate higher resolution videos; 2) train SynCamMaster with more viewpoints (e.g., >10) on larger memory GPUs.
>
> [1] Wu, Guanjun, et al. "4d gaussian splatting for real-time dynamic scene rendering." Proceedings of the IEEE/CVF Conference on Computer Vision and Pattern Recognition. 2024.
>
> >Robustness to out-of-distribution poses.
>
> Thank you for your valuable suggestion! We have added results on our project page (https://syncammaster.github.io/SynCamMaster/) with the input of a 3-meter difference in adjacent views, therefore view1 and view4 are 9 meters apart (greater than the maximum interval of 5.5 meters in the training data). The results demonstrate that our model has promising generalization capabilities, effectively generating views with significant visual differences, which meets the requirements of most real-world applications.
>
> >Evaluate the inter-view consistency by cycle consistency check.
>
> Thank you for the advice! Since SynCamMaster is designed to generate multi-view videos from text and camera poses simultaneously, we cannot generate some views first and then condition on the generated videos to create new views for a cycle consistency check.
> As an alternative, we assess 3D consistency in Tab. 1&2 by:
>
> 1. Using automatic metrics following previous work SV4D [1], including FVD-V and CLIP-V.
>
> 2. Calculating the number of matching pixels estimated by GIM [2] with confidence greater than the threshold, denoted as Mat. Pix.
>
> Additionally, we add qualitative results on 4D reconstruction on our project page.
>
> [1] Xie, Yiming, et al. "Sv4d: Dynamic 3d content generation with multi-frame and multi-view consistency." *arXiv preprint arXiv:2407.17470* (2024).
>
> [2] Shen, Xuelun, et al. "GIM: Learning Generalizable Image Matcher From Internet Videos." *arXiv preprint arXiv:2402.11095* (2024).
>
> >About dataset release.
>
> Yes, we will release the multi-view video data rendered by the UE engine.
>
> >How were the data mixture probabilities (0.6, 0.2, 0.2) chosen, and what role do they play in the model’s performance?
>
> Thanks for the advice! We added the analysis below in Appendix D:
>
> **Discussion on the Data Mixture Strategy** In the paper, we jointly train our model on multi-view video data, multi-view image data, and single-view video data with probabilities of 0.6, 0.2, and 0.2, respectively. We also explored the impact of different mixing ratios on the generated results. We found that a higher proportion of multi-view image data disrupts the temporal continuity of the videos. On the other hand, using too much single-view video data causes the model to favor synthesizing views with small relative angles, affecting the camera accuracy. Therefore, we sample multi-view image data and single-view video data with small probabilities.

---

> > ### Comment · Reviewer_kLvx · 2024-11-25
> >
> > Thanks for the new results.
> >
> > The 4D reconstruction looks not that good to me. Perhaps large motion in the video makes the rec harder, but I feel it is mainly due to the lack of cameras or multiview inconsistency in the generation

---

> ### Author Response · Authors · 2024-11-26
> **Further Response to Reviewer kLvx**
>
> Thanks for your feedback!
>
> Yes, the reconstruction quality is primarily affected by the sparsity of the views. Generating more views could further improve the 4D reconstruction results. **To verify this, we have updated the project page (https://syncammaster.github.io/SynCamMaster/) with the reconstruction results using 4-view videos from the Plenoptic Dataset [1].** Since the Plenoptic Dataset is recorded using synchronized camera setups to capture multi-view videos, it can be considered as "ground truth videos" to evaluate the performance of 4DGS [2] under sparse-view settings. Specifically, we first perform the center crop on the videos from the Plenoptic Dataset to match the aspect ratio of SynCamMaster. Then, we resized them to the same resolution as SynCamMaster to use as input for 4DGS. It can be observed the reconstruction results are competitive with reconstruction from videos generated with SynCamMaster, indicating that the performance is mainly limited by the number of available views. The reconstruction performance can be further improved by training SynCamMaster with larger memory GPUs to generate more views.
>
> Additionally, the 4D consistency of SynCamMaster outperforms baselines by a large margin. It is evidenced by the number of matching points in Tab. 1, where we utilize image matcher GIM [3] to calculate matching points for the same frame across different views and average them over the temporal dimension.
>
>
> [1] Li, Tianye, et al. "Neural 3d video synthesis from multi-view video." Proceedings of the IEEE/CVF Conference on Computer Vision and Pattern Recognition. 2022.
>
> [2] Wu, Guanjun, et al. "4d gaussian splatting for real-time dynamic scene rendering." Proceedings of the IEEE/CVF Conference on Computer Vision and Pattern Recognition. 2024.
>
> [3] Shen, Xuelun, et al. "GIM: Learning Generalizable Image Matcher From Internet Videos." ICLR 2024.

---

### Official Review · Reviewer_zv9N · 2024-11-04

**Soundness:** 2
**Presentation:** 3
**Contribution:** 2
**Rating:** 3
**Confidence:** 5

**Summary:**

This paper presents a multiview video dataset rendered using Unreal Engine. Additionally, it proposes a multiview video generative model architecture and training strategy to adapt a pretrained video diffusion model for simultaneous multiview generation.

**Strengths:**

(1) A multiview video dataset is presented.
(2) A multiview video diffusion model is trained using the proposed dataset.
(3) Comprehensive visualization results of the proposed method are provided.

**Weaknesses:**

(1) Encoding the camera pose matrix directly into the diffusion process as a condition is overly coarse. The authors should consider transforming camera poses into more explicit representations, such as ray positional encoding [1].
(2) Since the multi-view video is rendered rather than directly collected from real-world scenes, a significant concern is ensuring that the model learns only multi-view-related information rather than features specific to Unreal Engine-rendered models, materials, and textures. For instance, in Fig. 6 (a), how to avoid the output as the rendered scene? Authors should also introduce some real-world multiview videos to help balance the training of the diffusion model.
(3) The rotation angle in Fig. 6 (b) appears noticeably less than 90 degrees. Additionally, the precision of camera control is low, as indicated in Table 4.
(4) The proposed multi-view synchronization module is essentially a cross-attention layer, which lacks explicit modeling of 3D information.
(5) Given that camera poses are static in each rendered video under the proposed setup, it's unnecessary to render arbitrary camera poses. We can greatly lessen the burden of generative models and improve it accuracy by capturing and training more fewer camera poses, e.g., 6 views evenly distributed around 360 degrees, and applying 3D reconstruction on the generated views to get other different views.
[1] Gao, Ruiqi, et al. "Cat3d: Create anything in 3d with multi-view diffusion models." arXiv preprint arXiv:2405.10314 (2024).

**Questions:**

Please refer to the weakness section.

---

> ### Author Response · Authors · 2024-11-21
> **Response to Reviewer zv9N (1/2)**
>
> We thank the reviewer for the insightful comments, and we address the questions below:
>
> >Why not adopt more explicit camera pose representation, like ray positional encoding?
>
> Thanks for the advice! Following SPAD [1], we attempted to transform camera poses into ray positional encoding and input them into the camera encoder. However, we observed that there was no significant difference in model performance between the two representation methods:
>
> | Camera Representation | Roterr | Transerr |
> | :-------------------: | :----: | :------: |
> |   Plucker Embedding   |  0.12  |   0.60   |
> |  Extrinsic Embedding  |  0.12  |   0.58   |
>
> We also present the qualitative results in Fig. 10. As shown, there is no significant difference in the generated results between the two representations. We assume this is because the UE data (rendered multi-camera synchronized videos) has the same camera intrinsics, and in this case, both representations contain consistent information.
>
> **We add the results in Appendix D.**
>
> [1] Kant, Yash, et al. "SPAD: Spatially Aware Multi-View Diffusers." Proceedings of the IEEE/CVF Conference on Computer Vision and Pattern Recognition. 2024.
>
> >Explain the UE style issue of Fig. 6(a).
>
> **We believe there might be some misunderstandings regarding Fig 6(a).** In Fig 6(a), we used text prompts from the validation set, which correspond to the prompts for UE-rendered scenes. Therefore, the model outputs videos that are similar to the rendered scenes, which is correct.
>
> Regarding how we prevent the model from overfitting to UE rendering scene features, we: 1) incorporated more diverse real-world image (DL3DV-10K) and video (with stationary camera) data, as introduced in section 3.3; 2) during training, we froze the base T2V generation model and only trained the newly added modules to maximize the model's generalization ability. In our experiments, we found that these two designs effectively prevent the model from "only generating UE-rendered scenes" and enable it to generate multi-view videos in diverse scenes based on text prompts, as shown at https://syncammaster.github.io/SynCamMaster/.
>
>
> >The rotation angle in Fig. 6 (b) appears noticeably less than 90 degrees. Additionally, the precision of camera control is low, as indicated in Table 4.
>
> First, we would like to clarify that the first and second rows of Fig 6(b) denote the results of baseline methods that do not use diverse multi-view image data for training. So, these models exhibit poor generalization in open-domain settings, causing low accuracy of camera control (noticeably less than 90 degrees). In contrast, the third and fourth rows, as the results of models trained with diverse multi-view image data for joint training, demonstrated a substantial accuracy improvement compared to the first and second rows.
> Additionally, we have indeed found that generating videos with large angular changes is very challenging for the model, resulting in greater errors compared to generating videos with smaller relative pose differences. We believe this is primarily due to the limited scale of available multi-view video data (500 scenes, 36 cameras each), which is a challenge for this task.
>
> >The proposed multi-view synchronization module is essentially a cross-attention layer, which lacks explicit modeling of 3D information.
>
> Thank you for your suggestion. We attempted explicit modeling of 3D information by implementing epipolar attention in the view-attention layers. Specifically, when implementing epipolar attention, for the token at spatial position $(x, y)$ in view $i$, it only aggregates features from all tokens within view $i$ and tokens on the epipolar line in other views. In contrast, each token attends to all tokens in all views in the full attention setting. We found that although epipolar attention has low rotation error, it can result in inconsistency with the text prompt semantics (shown in Fig. 11) compared to the full-attention design. **The results are presented in Appendix D.**
>
> On the other, we found that it is sufficient to learn the spatial correspondence in a data-driven manner without explicit 3D modeling, which is consistent with the findings of recent work [1].
>
> [1] Jin, Haian, et al. "Lvsm: A large view synthesis model with minimal 3d inductive bias." *arXiv preprint arXiv:2410.17242* (2024).

---

> > ### Author Response · Authors · 2024-11-21
> > **Response to Reviewer zv9N (2/2)**
> >
> > >Advantage of our proposed task over the baseline: "generate fix view videos -> reconstruct 4D -> render other views" .
> >
> > The mentioned pipeline can also be used for generating "multi-view synchronized videos". However, our proposed approach offers several advantages in comparison:
> >
> > 1. Video quality is higher and can handle more scenes. Current work in the field of 4D reconstruction has the following issues: 1) The quality of new views decreases in real-world scenarios with visible artifacts, requiring many input views with high resolution; 2) Existing methods may fail in modeling large motions or dramatic scene changes [1].
> >
> > 2. Generation efficiency. Applying a diffusion model for end-to-end generation is much faster than the pipeline of first generating fixed views, then training 4D representations, and rendering novel views. In addition, the ‘generation-reconstruction-rendering’ pipeline requires per-scene training, while multi-view synchronized videos can be synthesized in a feed-forward pass of SynCamMaster.
> >
> > Overall, we believe the approach you mentioned can be more efficient if the goal is to obtain a 4D scene representation. However, if the aim is to apply the model to filmmaking, the approach in this paper has advantages in terms of video quality, generality across different scenes, and generation speed.
> >
> > [1] Wu, Guanjun, et al. "4d gaussian splatting for real-time dynamic scene rendering." *Proceedings of the IEEE/CVF Conference on Computer Vision and Pattern Recognition*. 2024.

---

> ### Author Response · Authors · 2024-11-29
> **Thanks for your review**
>
> Dear Reviewer zv9N:
>
> Thanks again for the constructive comments and the time you dedicate to the paper! Many improvements have been made according to your suggestions and other reviewers' comments. We also updated the manuscript and summarized the changes above. We hope that our responses above and the revision will address your concerns.
>
> Since the discussion is about to close (**Dec. 2nd**), we would be grateful if you would kindly let us know of any other concerns and if we could further assist in clarifying any other issues.
>
> Thanks a lot, and with sincerest best wishes
>
> Submission 1274 Authors

---

> > ### Author Response · Authors · 2024-12-02
> > **Friendly Reminder**
> >
> > Dear Reviewer zv6N:
> >
> > Thanks again for the constructive comments and the time you dedicate to the paper!
> >
> > **Since the discussion is about to close *today*, we would be grateful if you could let us know if we have addressed your concerns.**
> >
> > Thanks a lot, and with sincerest best wishes
> >
> > Submission 1274 Authors

---

### Official Review · Reviewer_bsPL · 2024-11-04

**Soundness:** 2
**Presentation:** 1
**Contribution:** 3
**Rating:** 6
**Confidence:** 3

**Summary:**

The paper "SynCamMaster: Synchronizing Multi-Camera Video Generation from Diverse Viewpoints" introduces a novel approach for generating synchronized multi-view videos from arbitrary viewpoints, addressing challenges in open-domain video generation that require consistent geometry and appearance across diverse camera angles. The authors leverage a pre-trained text-to-video model and enhance it with a multi-view synchronization module, designed to integrate inter-view feature consistency, which is achieved by incorporating a camera encoder and a unique attention mechanism. Additionally, they introduce a hybrid dataset and progressive training techniques to improve the generalization of the model to open-domain scenarios, effectively overcoming the scarcity of synchronized multi-view video data. Their experiments demonstrate superior cross-view synchronization and visual fidelity compared to baseline methods.

**Strengths:**

This paper introduces a forward-looking approach by achieving multi-camera real-world video generation, a pioneering advancement that is likely to drive significant progress in several areas in computer vision. Extensive experiments and ablation studies support the reliability of the proposed model, and the project website showcases video results that validate the method's effectiveness.

**Weaknesses:**

1.I believe the challenge identification section is unclear. Initially, the authors mention two challenges in this field, then state, "our research targets open-domain video generation from multiple arbitrary viewpoints, which has not been investigated." However, they do not clearly explain the reason for this research, nor do they clarify the limitations of existing methods and the higher demands that necessitate this research.
2.The explanation of the model structure and model formulation is severely lacking, making it difficult to connect the input-to-output process in the model with the text description, including the loss function and optimization mechanism. Specifically, the figure should indicate what the input is, which step produces the features, and how the loss is calculated, among other key details.
3.How does SynCamMaster determine the rotation center and radius when generating rotating camera viewpoints? I think that for a method designed to generate multi-view videos, this is an aspect that requires clarification.
4. Although the paper provides results for camera rotations like 90 degrees or 20 degrees, how can you quantitatively assess that the generated images match the rotation you specified? Without ground truth for reference, it is challenging to determine whether the rotation results are accurate. Could the authors use a COLMAP or similar method to register the spatial relationships of the generated multi-view images in camera space?

**Questions:**

1.The citations for each method in tables should be added.
2.It appears a new synthetic dataset was proposed in this work for training the model. Will you decide to release this dataset? The dataset could be considered a contribution to the paper, so I suggest naming this synthetic dataset for readability.
3.In the qualitative experiments, you mention a polar bear case, but Figure 5 uses an elephant and a bus, and I couldn’t find polar bear images in the appendix.
4.In the quantitative experiments, your method performs worse than I2V-ours in terms of visual quality. I would like to see some analysis on this comparison between your results and those of I2V-ours.
5.Table 1 entries for I2V and SVD-XT should be unified if they refer to the same method for consistency.
6.In Figure 6(a), joint training strategy ablation study results are shown, but without ground truth to assess and with images that are too small, it is difficult to judge if the strategy improved the generation results, or if the effect is simply not significant. In contrast, Figure 6(b) shows clear improvements.

**Details Of Ethics Concerns:**

In the paper, authors generate some human (little girl) images through the diffusion model, I'm not sure whether it has ethics problem.

---

> ### Author Response · Authors · 2024-11-21
> **Response to Reviewer bsPL (1/2)**
>
> Thanks for your constructive comments! We address the concerns below:
>
> >The challenge identification part is unclear, especially in clarifying the limitation of existing methods.
>
> Thanks for the constructive comments. We have revised the related parts (line:68~78) in the introduction section to clarify the limitations of existing methods, and we believe the flow and logic of the description get better. We also introduce the necessitate of the research below.
>
> Our task of generating multi-camera videos from diverse viewpoints is an under-explored problem that also enables some unique applications.
> 1. **Film Production**: In real-world filmmaking, switching back and forth between multiple angles is a commonly used technique to create a storytelling atmosphere. For example, in a conversation between two people, the camera might switch from focusing on one speaker to a wide shot of the entire scene, maintaining temporal continuity. Generating synchronized multi-view videos can meet this need. For instance, a creator can splice together the video from view_i during the time period t_0 to t_1 with the video from view_i+1 during the time period t_1 to t_2 to create the desired sequence. In contrast, monocular video with a dynamic camera can not achieve this effect because a single camera cannot instantly switch between two vastly different perspectives.
> 2. **Used as Training Data for Downstream Tasks**: Synchronized multi-view video generation can be used as training data for various downstream tasks. For example, understanding a scene from multiple viewpoints is often necessary for visual robotic manipulation [1, 2, 3]. Additionally, in 3D human pose estimation [4, 5, 6], multiple viewpoints are required to accurately estimate the 3D posture of a person.
> To this end, we designed SynCamMaster as an exploration of using diffusion models for open-domain synchronized multi-view video generation.
>
> [1] Akkaya, Ilge, et al. "Solving rubik's cube with a robot hand." arXiv preprint arXiv:1910.07113 (2019).
>
> [2] James, Stephen, et al. "Coarse-to-fine q-attention: Efficient learning for visual robotic manipulation via discretisation." CVPR 2022.
>
> [3] Seo, Younggyo, et al. "Multi-view masked world models for visual robotic manipulation." ICML, 2023.
>
> [4] Wang, Jinbao, et al. "Deep 3D human pose estimation: A review." Computer Vision and Image Understanding, 2021.
>
> [5] Dong, Junting, et al. "Fast and robust multi-person 3d pose estimation from multiple views." CVPR 2019.
>
> [6] Mitra, Rahul, et al. "Multiview-consistent semi-supervised learning for 3d human pose estimation." CVPR 2020.
>
>
> >Unclear exposition about the model formulation and the input/output of some key components.
>
> Thank you for your reminder! We have made the following updates:
>
> 1. Added a more detailed illustration figure and corresponding introduction of the base text-to-video generation model in Appendix A.
> 2. Modified the pipeline figure (Fig. 2) in the main text to highlight the correspondence between Fig. 2(a) and Fig. 2(b).
> 3. Introduced the loss function and optimization mechanism in Section 3.1.
>
> > How does SynCamMaster determine the rotation center and radius when generating rotating camera viewpoints?
>
> SynCamMaster is a multi-camera video generation model that uses text prompts and camera poses as conditions. If we want to generate videos from rotating camera viewpoints, we only need to define the camera poses (R,T) of all viewpoints in such a configuration and feed them into the model as conditions. More specifically, we regard the camera coordinate system of the first camera as the global system, where we first define a circle with a specified center and radius, then we can place the cameras on the circle by specifying their translation vector T and ensure all the cameras orient to the center by specifying their pose matrix R.
>
> > Evaluate the control accuracy by estimating poses from the generated videos and comparing with the given poses.
>
> Thank you for your valuable suggestion! We use the state-of-the-art image matcher GIM [1] to obtain the estimated pose and evaluate the camera accuracy in our paper (introduced in lines 370-373), and the results are shown in Table 4. The reason we do not use COLMAP is we found that COLMAP typically requires more cameras and higher image resolution to complete the registration. Even when we used ground truth images rendered by Unreal Engine with 4 cameras at a resolution of 384x672, the registration often failed. Therefore, we chose GIM as an alternative.
>
> [1] Shen, Xuelun, et al. "GIM: Learning Generalizable Image Matcher From Internet Videos." *arXiv preprint arXiv:2402.11095* (2024).

---

> > ### Author Response · Authors · 2024-11-21
> > **Response to Reviewer bsPL (2/2)**
> >
> > >The citations for each method in tables should be added.
> >
> > Thanks for the suggestion! We do not add citations in the table because using the "author, year" format would make the table too wide to fit the layout. We ensure that the baseline methods are cited and described in detail in the paragraph introducing the baselines (lines 404-415).
> >
> > >About releasing our multi-camera UE dataset.
> >
> > Yes, we will release the multi-view video data rendered by the UE engine. In the main paper, we uniformly refer to the Unreal Engine-rendered multi-view video data as "M. V. Video".
> >
> > >In the qualitative experiments, you mention a polar bear case, but Figure 5 uses an elephant and a bus, and I couldn’t find polar bear images in the appendix.
> >
> > Sorry for the confusion. We have corrected the corresponding text description in line 426, thanks for pointing it out.
> >
> > >In the quantitative experiments, your method performs worse than I2V-ours in terms of visual quality. I would like to see some analysis on this comparison between your results and those of I2V-ours.
> >
> > We add an analysis of the differences in visual effects between the proposed "multi-camera synchronized video generation" method (SynCamMaster) and our trained I2V generation model (I2V-ours) as follows:
> >
> > We observed that SynCamMaster underperforms our trained I2V generation model in terms of the visual effects of the generated videos. The main reason lies in the quantity and diversity of the training data. Specifically, the data used to train I2V consists of ~100K collected high-quality videos, whereas the multi-view videos used to train SynCamMaster amount to only 18K. Additionally, the I2V training data covers various scenes from daily life, while the rendered multi-view videos are primarily of urban street scenes, which lack diversity. Therefore, despite using DL3DV-10K and monocular videos for regularization, the visual quality is still slightly inferior to that of the trained I2V model.
> >
> >
> > >Table 1 entries for I2V and SVD-XT should be unified if they refer to the same method for consistency.
> >
> > In Table 1, I2V-Ours and SVD-XT refer to different methods. I2V-Ours refers to the image-to-video (I2V) generation model that we trained based on the same T2V generation model as SynCamMaster, as introduced in Lines 410-415. SVD-XT, on the other hand, is the I2V generation model proposed in [1].
> >
> > [1] Blattmann, Andreas, et al. "Stable video diffusion: Scaling latent video diffusion models to large datasets." *arXiv preprint arXiv:2311.15127* (2023).
> >
> > >In Figure 6(a), joint training strategy ablation study results are shown, but without ground truth to assess and with images that are too small, it is difficult to judge if the strategy improved the generation results, or if the effect is simply not significant. In contrast, Figure 6(b) shows clear improvements.
> >
> > We would like to clarify that in Fig. 6(a), there is no significant difference in the generated results of the four training methods. This is because Fig. 6(a) shows the results on the validation set, which is consistent with the training scene domain. Training solely on the rendered data can achieve good results on the validation set. The main differences between the different training strategies lie in their performance on other domains (which we refer to as the test set), and these results are shown in Fig. 6(b).

---

> > ### Comment · Reviewer_bsPL · 2024-11-25
> >
> > I’m very pleased to see that you have carefully addressed and appropriately revised every question I raised. However, I still have one remaining question regarding the definition of the rotation center and radius in SynCamMaster.
> >
> > From your response, I understand that the input to the model only includes the camera pose (R, T). The camera position in the global system, the rotation center and radius are determined by the model itself based on the provided camera pose. Wouldn't this lead to significantly different results for multiple outputs generated from the same video? Considering the inherent uncertainty of diffusion models, this aspect raises some concerns for me. One situtation comes to my mind. If you use a video to generate a multi-view video, for example, you rotate the camera pose by 90 degrees. And then you use this generated video as input to rotate it back to the original angle, the resulting viewpoints might not match the original video due to differences in the rotation radius or center.

---

> ### Author Response · Authors · 2024-11-25
> **Further Response to Reviewer bsPL**
>
> Dear Reviewer bsPL,
>
> Thanks again for your elaborate review and the prompt reply! We address your concerns below:
>
> We would like to explain that the rotation center and radius are not determined by the model itself, but are uniquely defined by the relative extrinsic matrix $ [R|T] $ of the camera. We provide a detailed explanation using an example of two cameras positioned on a circular arc, both facing the center of the circle, with an azimuth angle difference of $n$ degrees:
>
> Firstly, according to the principles of imaging, the rotation center lies on the principal ray (the ray that passes through the camera's optical center and the principal point on the image plane). Secondly, for any two camera positions with an azimuth change of $n$ degrees, we can derive the rotation radius $r$ from their relative translation vector $T$ using the formula $r$=norm(T) / 2 / sin(n/2). Therefore, the rotation radius is uniquely determined by the relative pose $ [R|T] $. In other words, given the same video input and the same relative pose, the model should output a video with a consistent rotation radius.
>
> To verify this, we conducted the following two sets of experiments:
>
> 1. We performed two inferences with SynCamMaster using the same input reference video, relative extrinsic matrix, and text prompt, but with different random seeds. The results are shown in Figure A on the project page (https://syncammaster.github.io/SynCamMaster/#Fig_A, at the bottom of the page if the link doesn't jump). It can be observed that the output variations due to different random seeds have consistent relative angles, with differences only in the outpainting areas where the new view does not overlap with the input view.
>
> 2. Given the relative extrinsic matrix $[R|T]$, input reference video $V_i$, and text prompt $P_t$, we used SynCamMaster to generate a new view video $V_o = $SynCamMaster$(V_i, [R|T], P_t)$. Then, using the generated new view video as input, we generated a video with $[R|T]^{-1}$ and $P_t$ as conditions, i.e., $V_{i}' = $SynCamMaster$(V_o, [R|T]^{-1}, P_t)$. The results are shown in Figure B (https://syncammaster.github.io/SynCamMaster/#Fig_B). It can be observed that the final output $V_{i}'$ is almost identical to the input reference video $V_i$ in terms of viewpoint, which verifies our explanation.
>
> We hope we have correctly understood your concerns and addressed them. We would be grateful if you would kindly let us know of any other concerns and if we could further assist in clarifying any other issues.

---

### Official Review · Reviewer_KEwM · 2024-11-07

**Soundness:** 3
**Presentation:** 3
**Contribution:** 3
**Rating:** 8
**Confidence:** 4

**Summary:**

This paper introduces SynCamMaster, a pioneering framework for generating real-world videos across multiple camera perspectives. SynCamMaster also extends to novel view synthesis, allowing re-rendering of input videos from new viewpoints. To address the scarcity of multi-camera video data, the authors propose a hybrid data construction and training paradigm that enhances the model's generalization capabilities. Extensive experiments demonstrate that SynCamMaster significantly outperforms baseline methods, with particularly impressive visualization results.

**Strengths:**

1. Simplicity and Efficiency: The proposed method is both straightforward and efficient. This paper introduces a multi-view synchronization module designed to ensure camera-conditioned video consistency, enhancing realism and alignment across views.

2. Innovative Data Construction: This paper addresses the scarcity of multi-view video data by developing a robust hybrid-data collection pipeline. This process effectively collects diverse and comprehensive training data.

3. Thoughtful Training Strategy: The paper provides a clear breakdown of each stage in the training strategy, along with detailed insights into the motivations behind each stage. These insights offer readers a better understanding of the model's optimization process and rationale.

4. High-Quality Visualizations: The visualization results presented in the paper are impressive.

5. Clarity of Presentation: The paper is well-organized and articulated in a way that is accessible and easy to understand.

**Weaknesses:**

Lack of Comparison with MotionCtrl: A drawback is the absence of a comparative evaluation with MotionCtrl ("MotionCtrl: A Unified and Flexible Motion Controller for Video Generation").

**Questions:**

See weaknesses.

---

> ### Author Response · Authors · 2024-11-21
> **Response to Reviewer KEwM**
>
> Thanks for providing encouraging comments on our paper! We are encouraged to see that SynCamMaster is acknowledged to present an effective approach to multi-camera video generation, with innovative data construction and training strategy, as well as high-quality visual results. We address your concerns below:
>
> >Lack of Comparison with MotionCtrl.
>
> Thanks for the suggestion! We add qualitative comparisons with MotionCtrl on our project page (https://syncammaster.github.io/SynCamMaster/).

---

> > ### Comment · Reviewer_KEwM · 2024-11-28
> >
> > Thanks for the project page. Any comments, explanation or conclusion from the comparison?

---

> > > ### Author Response · Authors · 2024-11-29
> > > **Further Response to Reviewer KEwM**
> > >
> > > Dear Reviewer KEwM,
> > >
> > > Thanks for your reply! We supplement the following description, analysis, and conclusion regarding the comparison:
> > >
> > > We compared our method with the state-of-the-art single-video camera control approach, MotionCtrl [1]. Specifically, we utilized SynCamMaster to synthesize multi-view images (M.V. Images) and used them as the image input to the SVD [2] version of MotionCtrl. Since MotionCtrl is designed to generate monocular videos that adhere to the input camera pose, it could not effectively ensure synchronization and content consistency across multiple videos due to the lack of cross-view feature aggregation. For example, the bus remains almost stationary in the first view while accelerating forward in the second view. In contrast, thanks to the designed multi-view synchronization module and training on multi-view videos, SynCamMaster can generate view-synchronized videos that adhere to both the camera poses and text prompts.
> > >
> > > To summarize, MotionCtrl is elaborately designed to generate monocular videos with camera motion control, while SynCamMaster performs better in generating synchronized multi-view videos. Additionally, we believe that effectively combining MotionCtrl and SynCamMaster to generate synchronized multi-view videos with camera motion control in each view is worth exploring, which we leave as future work.
> > >
> > > [1] Wang, Zhouxia, et al. "Motionctrl: A unified and flexible motion controller for video generation." *ACM SIGGRAPH 2024 Conference Papers*. 2024.
> > >
> > > [2] Blattmann, Andreas, et al. "Stable video diffusion: Scaling latent video diffusion models to large datasets." *arXiv preprint arXiv:2311.15127* (2023).

---

### Official Review · Reviewer_bc9Q · 2024-11-08

**Soundness:** 3
**Presentation:** 3
**Contribution:** 3
**Rating:** 6
**Confidence:** 4

**Summary:**

This paper presents a multi-view video generation approach based on an input textual prompt or monocular video. To achieve this goal, a multi-view synchronization module is incorporated into a pre-trained text-to-video model, where the goal is to maintain appearance and geometry consistency across multiple viewpoints. To address the scarcity issue of the training data, a progressive training scheme is designed that leverages multi-camera images, synthetic multi-view videos, and real-world monocular videos.

**Strengths:**

1. The multi-view video generation is a novel task. This paper shows some promising results in this direction, where the temporal consistency within each single view and spatial consistency across different views are maintained reasonably well.

2. The proposed way of using multi-view images, synthetic multi-view videos, and real-world monocular videos to train the model is novel. Ideally, we could train the model with multi-view videos, which are however hard to obtain, especially the high-quality real-world ones. The proposed approach shows an alternative approach to solve the data scarcity issue.

**Weaknesses:**

1. The proposed approach can only generate a single video at a novel viewpoint each time. Compared with generating multiple videos at different viewpoints simultaneously, the view consistency among the generated multi-view videos may not be good. Of course, the proposed approach can be used progressively generate further and further novel-view videos. But the view inconsistency may accumulate.

2. I wish the authors could show more visual results so the quality of the generated videos could be more comprehensively gauged. In the current results (both in the paper and on the project webpage), the same example (e.g., the gorilla eating carrot, the elephant walking on a beach) appears many times.

**Questions:**

1. Around the line #189, should $\phi_t$ be $\psi_t$? And what do $\psi_t^{-1}$ and $\psi_t'$ mean, respectively?

2. Could you please provide more information about the real-world monocular videos used in this paper? Like duration, amount, how they are curated, etc.

3. In the line #140, does $f$ indicate the number of frames?

4. Is there any plan to release the code?

5. Can more visual examples of the generated multi-view synthetic videos be shown?

---

> ### Author Response · Authors · 2024-11-21
> **Response to Reviewer bc9Q**
>
> We thank the reviewer for providing encouraging comments on our paper. We provide clarifications to the concerns below:
>
> >The proposed approach can only generate a single video at a novel viewpoint each time.
>
> We would like to claim that SynCamMaser is designed to generate multiple videos from different viewpoints simultaneously, rather than generate one video each time. Specifically, during inference, we denoise $n$ noisy latents simultaneously, and their features interact through the designed multi-view synchronization module, resulting in the simultaneous output of $n$ videos. In our paper, we validated the effectiveness of our method under three settings of $n=2, 4$. We believe it meets the needs of most practical applications and can further increase $n$ if sufficient GPU memory is available.
>
> >Show more visual results of different cases.
>
> Thank you for the advice! We have updated the project page (https://syncammaster.github.io/SynCamMaster/) with more video results in the **Demos** and **More Results** sections.
>
> >Around the line #189, should $\phi_t$ be $\psi_t$? And what do $\psi_t^{-1}$ and $\psi_t'$ mean, respectively?
>
> Yes, $\phi_t$ should be $\psi_t$ in line 189, we have corrected the typo in the main text, thanks for pointing it out! $\psi_t^{-1}$ represents the inverse function of $\psi_t$, and $\psi_t'$ represents the first derivative of $\psi_t$.
>
> >More details about the real-world monocular videos used in this paper.
>
> Thank you for your suggestion! We introduce the construction process of real-world monocular videos in lines 296-303 of the main text, accompanied by a detailed description in lines 869-879 of the appendix.
>
> >In the line #140, does $f$ indicate the number of frames?
>
> Yes, we add the explanation in line 142.
>
> >Is there any plan to release the code?
>
> Yes, we have transferred the proposed method to the open-source video generation models CogVideoX-2B and CogVideoX-5B [1],  and the performance is generally comparable to that in our paper. We will open-source the code implemented on these CogVideoX models and release the multi-view video data rendered by the UE engine used in the paper.
>
> [1] Yang, Zhuoyi, et al. "Cogvideox: Text-to-video diffusion models with an expert transformer." *arXiv preprint arXiv:2408.06072* (2024).

---

### Official Review · Reviewer_edwQ · 2024-11-08

**Soundness:** 3
**Presentation:** 3
**Contribution:** 2
**Rating:** 6
**Confidence:** 4

**Summary:**

This paper focuses on the multi-view video generation with image/video and camera pose  conditions. Specifically, this paper propose a plug-and-play multi-view synchronization module to maintain appearance and geometry consistency. In addition, they collect a small set of 4D data and combine it together with 3D data to improve the generalization ability.

**Strengths:**

1. The method is novel and each module and design are reasonable with clear motivation.

2. The proposed method achieved good performance both qualitatively and quantitatively.

3. Extensive experiments and ablation studies demonstrate the effectiveness of the proposed techniques.

**Weaknesses:**

1. I am kind of confused with the motivation of this task. The multi-camera videos generated by this work are several videos with fixed cameras. However, the videos we see and use in our daily life are commonly one monocular video with a dynamic camera, like the task that MotionCtrl focuses on. I hope the authors could provide more explanation about the task motivation.

2. As claimed in the introduction section, multi-camera synchronized video generation is crucial for virtual filming. I am not an expert in virtual filming, but I feel it requires highly spatio-temporal consistent videos. Therefore, a good way to demonstrate the 4D consistency is to lift the generated samples to 4D representations, such as 4D gaussian splatting, and to show the rendered views. Plenoptic dataset [A] can be used for this evaluation.


3. Evaluation data and metrics should be explained. The details of the evaluation data are not provided. It is more convincing to evaluate on the public dataset. For multi-camera video evaluation, the PSNR, SSIM, and LPIPS metrics on Plenoptic [A]  D-NeRF [B] datasets can be used. For video quality evaluation, VBench [C] would be better than ``100 manually collected text prompts’’. In addition, when evaluating the accuracy of camera control with TansErr, the estimated camera should be normalized before estimating the loss (following [D]), since the SfM is up to a scale and the scale mismatch can greatly increase the TransErr. In CameraCtrl, the scale problem is not significant as the compared methods are all trained on the Re10K dataset. However, this paper trains on a different dataset (DL3DV-10K) with MotionCtrl (Re10K) and the scale problem might be enhanced.

*Reference*

[A] Neural 3D Video Synthesis from Multi-view Video

[B] D-NeRF: Neural Radiance Fields for Dynamic Scenes

[C] VBench: Comprehensive Benchmark Suite for Video Generative Models

[D] CamCo: Camera-Controllable 3D-Consistent Image-to-Video Generation

**Questions:**

My concern is mainly about the evaluation of multi-camera videos. Please refer to Weaknesses.

---

> ### Author Response · Authors · 2024-11-21
> **Response to Reviewer edwQ (1/2)**
>
> We thank the reviewer for the detailed feedback on our paper!
>
> >Clarification on the motivation of our task.
>
> Our task of generating multi-camera videos from diverse viewpoints is an under-explored problem that also enables some unique applications.
> 1. **Film Production**: In real-world filmmaking, switching back and forth between multiple angles is a commonly used technique to create a storytelling atmosphere. For example, in a conversation between two people, the camera might switch from focusing on one speaker to a wide shot of the entire scene, maintaining temporal continuity. Generating synchronized multi-view videos can meet this need. For instance, a creator can splice together the video from view_i during the time period t_0 to t_1 with the video from view_i+1 during the time period t_1 to t_2 to create the desired sequence. In contrast, monocular video with a dynamic camera can not achieve this effect because a single camera cannot instantly switch between two vastly different perspectives.
> 2. **Used as Training Data for Downstream Tasks**: Synchronized multi-view video generation can be used as training data for various downstream tasks. For example, understanding a scene from multiple viewpoints is often necessary for visual robotic manipulation [1, 2, 3]. Additionally, in 3D human pose estimation [4, 5, 6], multiple viewpoints are required to accurately estimate the 3D posture of a person.
> To this end, we designed SynCamMaster as an exploration of using diffusion models for open-domain synchronized multi-view video generation.
>
> [1] Akkaya, Ilge, et al. "Solving rubik's cube with a robot hand." *arXiv preprint arXiv:1910.07113* (2019).
>
> [2] James, Stephen, et al. "Coarse-to-fine q-attention: Efficient learning for visual robotic manipulation via discretisation." CVPR 2022.
>
> [3] Seo, Younggyo, et al. "Multi-view masked world models for visual robotic manipulation." ICML, 2023.
>
> [4] Wang, Jinbao, et al. "Deep 3D human pose estimation: A review." *Computer Vision and Image Understanding,* 2021.
>
> [5] Dong, Junting, et al. "Fast and robust multi-person 3d pose estimation from multiple views." CVPR 2019.
>
> [6] Mitra, Rahul, et al. "Multiview-consistent semi-supervised learning for 3d human pose estimation." CVPR 2020.
>
> > Evaluating 4D consistency of our results.
>
> Thanks for the advice! We add the 4D reconstruction results with 4DGS [1] to our project page (https://syncammaster.github.io/SynCamMaster/). Specifically, we condition SynCamMaster on four cameras with 10&deg; difference in azimuth and the text prompt to synthesize multi-view videos. Then, we use the four view camera parameters and our generated videos as input to 4DGS [1], and follow their open-source code (https://github.com/hustvl/4DGaussians) to train the 4D Gaussian splatting. As exhibited, the rendered novel views are overall consistent with fewer artifacts.
>
> It's worth noting that it's quite challenging to reconstruct 4D with sparse views (4 views in the case) at low resolution (384x672 in the case). To verify this, we have updated the project page (https://syncammaster.github.io/SynCamMaster/) with the reconstruction results using 4-view videos from the Plenoptic Dataset. Since the Plenoptic Dataset is recorded using synchronized camera setups to capture multi-view videos, it can be considered as "ground truth videos" to evaluate the performance of 4DGS [1] under sparse-view settings. Specifically, we first perform the center crop on the videos from the Plenoptic Dataset to match the aspect ratio of SynCamMaster. Then, we resized them to the same resolution as SynCamMaster to use as input for 4DGS. It can be observed the reconstruction results are competitive with reconstruction from videos generated with SynCamMaster, indicating that the performance is mainly limited by the number of available views. The reconstruction performance can be further improved by training SynCamMaster with larger memory GPUs to generate more views.
>
> On the other, the 4D consistency of SynCamMaster outperforms baselines by a large margin. It is evidenced by the number of matching points in Tab. 1, where we utilize image matcher GIM [2] to calculate matching points for the same frame across different views and average them over the temporal dimension.
>
> [1] Wu, Guanjun, et al. "4d gaussian splatting for real-time dynamic scene rendering." CVPR 2024.
>
> [2] Shen, Xuelun, et al. "GIM: Learning Generalizable Image Matcher From Internet Videos." ICLR 2024.
>
> >Evaluation data and metrics should be explained.
>
> For the evaluation data, we use prompts synthesized by our finetuned LLM across various scenarios. We will open-source these prompts to facilitate future comparative research. Additionally, we include experiments using the open-source VBench prompts, and the results are exhibited in Tab. 7 in Appendix E. For the evaluation metrics, we add a detailed introduction to the metrics in Section 4.1.

---

> > ### Author Response · Authors · 2024-11-21
> > **Response to Reviewer edwQ (2/2)**
> >
> > >About using PSNR, SSIM, and LPIPS metrics on Plenoptic, D-NeRF datasets.
> >
> > Since our model is designed to synthesize synchronized multi-camera videos from **text** and camera poses, we cannot effectively use reconstruction metrics such as PSNR and SSIM to evaluate its quality because no ground truth is available. As an alternative, we assess the visual quality with FID, FVD, and CLIP scores, and evaluate 3D consistency by 1) using automatic metrics following previous work SV4D [1], including FVD-V and CLIP-V; 2) calculating the number of matching pixels estimated by GIM [2] with confidence greater than the threshold, denoted as Mat. Pix.
> >
> > [1] Xie, Yiming, et al. "Sv4d: Dynamic 3d content generation with multi-frame and multi-view consistency." *arXiv preprint arXiv:2407.17470* (2024).
> >
> > [2] Shen, Xuelun, et al. "GIM: Learning Generalizable Image Matcher From Internet Videos." *arXiv preprint arXiv:2402.11095* (2024).
> >
> > >Results on VBench.
> >
> > We randomly sample 100 prompts from the 300 prompts provided by VBench in the categories of 'animal', 'human', and 'vehicles' for evaluation. We do not sample from categories such as 'plant' and 'scenery' because, given that we generate videos from fixed camera positions, static scenes would result in videos that are stationary over time. The evaluation results are presented in Tab. 7 in Appendix D. Our method outperforms the baseline methods across all metric dimensions and demonstrates performance comparable to our trained I2V model.
> >
> > | Method       | Subject Consistency | Background Consistency | Aesthetic Quality | Imaging Quality | Temporal Flickering | Motion Smoothness |
> > | ------------ | ------------------- | ---------------------- | ----------------- | --------------- | ------------------- | ----------------- |
> > | SVD-XT       | 94.32               | 94.23                  | 48.85             | 52.80           | 95.79               | 98.26             |
> > | CameraCtrl   | 95.91               | 96.40                  | 50.44             | 52.85           | 96.79               | 98.73             |
> > | I2V-Ours     | 93.53               | 92.93                  | 49.07             | 58.49           | 95.20               | 98.13             |
> > | SynCamMaster | 97.84               | 96.55                  | 50.50             | 58.30           | 98.95               | 99.27             |
> >
> > >Solving the scale problem.
> >
> > Thank you for your reminder! We normalize the translation vector following CamCo [1], update the data in Table 3, and add the corresponding explanation in lines 464-469 of the main text. Note that since GIM is an image pair matching algorithm, we normalize the relative translation vector between any two views to have an L_2 norm of 1.0, rather than normalizing based on the two views with the greatest difference. We also include a description of the construction of the camera accuracy evaluation set in Appendix C.
> >
> > [1] Xu, Dejia, et al. "CamCo: Camera-Controllable 3D-Consistent Image-to-Video Generation." *arXiv preprint arXiv:2406.02509* (2024).

---

> > > ### Comment · Reviewer_edwQ · 2024-11-28
> > >
> > > Thanks authors for the response and providing additional experiments.
> > >
> > > Most of my concerns are addressed and I would like to increase my rating.

---

> > > > ### Author Response · Authors · 2024-11-28
> > > > **Further Response to Reviewer edwQ**
> > > >
> > > > Thanks for the reply and appreciation of our work! We are committed to continuously improving our work and will incorporate the results into the final version.

---

### Official Review · Reviewer_zZFD · 2024-11-09

**Soundness:** 3
**Presentation:** 2
**Contribution:** 2
**Rating:** 6
**Confidence:** 4

**Summary:**

In this manuscript, the authors proposed a plug-and-play module that enhances a pretrained text-to-video model for multi-camera video generation. To achieve this goal, a multi-view synchronization module to ensure appearance and geometry consistency across different viewpoints. The synthesized videos using UE are generated as supplement for training. The proposed SynCamMaster is compared with related works on multi-view video generation, and provides competitive performance.

**Strengths:**

The proposed method is simple yet effective for multi-camera video generation.
The paper is well-written and easy to understand.
The results are impressive.

**Weaknesses:**

[Pioneer Status] In terms of pioneering work, it has come to my attention that Sora possesses multi-camera video generation capabilities. Therefore, the claim in the article that "To our knowledge, SynCamMaster pioneered multi-camera real-world video generation" warrants reconsideration.

[Novelty of SynCamMaster] The core design of SynCamMaster revolves around the inter-view synchronization module, which can be integrated into a pretrained text-to-video model. However, similar modules are commonly found in other generation models employing cross-view attention, which suggests that the novelty is incremental. It would be beneficial to emphasize how SynCamMaster distinguishes itself from existing methods.

[Camera Encoder] The process for converting 12-dimensional extrinsic parameters to align with the dimensions of spatial features remains unclear. The encoding of camera extrinsics significantly influences the final outcome, so providing additional crucial details would be appreciated.

[Typos] M.V. Images in table 1 and 4, they are mentioned in the manuscript. It is better to make it clear what is it.

[Supplementary files] If the generated video files by all comparison methods are provided as supplementary files, it will provide a better and more intuitive way to understand the proposed SynCamMaster.

**Questions:**

See Weaknesses.

---

> ### Author Response · Authors · 2024-11-21
> **Response to Reviewer zZFD**
>
> We thank the reviewer for the elaborate review! We will address your concerns below:
>
>
> >Response to the comment that Sora possesses multi-camera video generation capabilities.
>
>
> We have not found any indication in Sora's technical report that it has the capability to generate "multi-view synchronized videos". We would greatly appreciate it if you could point us to any documentation or information that demonstrates Sora's ability in this regard.
>
> >Emphasize how SynCamMaster distinguishes itself from existing methods since the inter-view synchronization module is similarly designed.
>
> As you mentioned, SynCamMaster has a similar module design with some methods in related fields. However, we would like to emphasize our distinct contributions:
>
> 1. We explore a new task, specifically the generation of open-domain multi-view synchronized videos.
> 2. We identify the importance of dataset construction for this task and propose a mixed dataset composed of multi-view images, multi-view videos, and general videos, as illustrated in Fig. 3.
> 3. We design a progressive training strategy with gradually increasing relative angles between views and employ joint training to enhance generalization and video quality.
>
> >Details about how the 12 extrinsic parameters are processed to align with the spatial feature.
>
> Thanks for the advice! In each block, we employ a MLP (namely the camera encoder) to project the 12 parameters into feature embedding of $d$ dimension (the spatial features dimension). The corresponding description can be found in lines 214-215 in the main text.
>
> >What is M.V. Images in Tables 1 and 4.
>
> Thanks for pointing it out! It refers to the multi-view image generated by SynCamMaster. We add the corresponding explanation at lines 423-425 in the main text.
>
> > Provide the video comparison with existing methods.
>
> Thank you for your reminder. We have added video results comparing SynCamMaster with baseline methods on the project page (https://syncammaster.github.io/SynCamMaster/).

---

### Author Response · Authors · 2024-11-21
**Summary of Revision**

We thank all the reviewers for your time and thoughtful feedback. We revised our manuscript and submitted a new version for review. We also add more video results on our anonymous project page (https://syncammaster.github.io/SynCamMaster/) for your reference.

## Summary of Revision
### Project Page
1. Including more synthesized results of SynCamMaster in "Demos" and "More Results" (Reviewer bc9Q).
2. Adding video comparison of SynCamMaster and baseline methods including MotionCtrl [1] (Reviewer zZFD, KEwM).
3. Results on 4D reconstruction (Reviewer edwQ, kLvx).
4. Results with Out-of-Distribution Cameras (Reviewer kLvx).
### Main Paper
1. We revise the introduction section to better clarify the limitations of existing methods (Reviewer bsPL).
2. We refine the method pipeline figure for a clear correspondence with the text (Reviewer bsPL).
3. We add the introduction of the camera encoder (Reviewer zZFD).
4. We normalize the translation vector following [2], update the results in Table 3, and add the corresponding explanation (Reviewer edwQ).
5. We fix some typos.
### Appendix
1. Detailed introduction of the base text-to-video generation model in Appendix A (Reviewers bsPL).
2. Description of the evaluation set construction in Appendix C (Reviewers edwQ).
3. Comparison of using different camera representations in Appendix D (Reviewers zv9N).
4. Results of using epipolar attention as explicit 3D modeling in Appendix D (Reviewers zv9N).
5. Discussion on the data mixture strategy in Appendix D (Reviewers kLvx).
6. Comparison with baseline methods on VBench in Appendix E (Reviewers edwQ).

[1] Wang, Zhouxia, et al. "Motionctrl: A unified and flexible motion controller for video generation." ACM SIGGRAPH 2024 Conference Papers. 2024.

[2] Xu, Dejia, et al. "CamCo: Camera-Controllable 3D-Consistent Image-to-Video Generation." arXiv preprint arXiv:2406.02509 (2024).

---

### Author Response · Authors · 2024-11-25
**Thanks for Your Review**

Dear Reviewers,

Thanks again for the constructive comments and the time you dedicate to the paper! Many improvements have been made according to your suggestions and other reviewers' comments. We also updated the manuscript and summarized the changes below. We hope that our responses and the revision will address your concerns.

Since the discussion is about to close, we would be grateful if you could take some time to review our responses and the revised manuscript. We are glad to follow up with your further comments.

Thanks a lot again, and with sincerest best wishes

Submission 1274 Authors

---

### Author Response · Authors · 2024-11-29
**Friendly Reminder**

Dear Reviewers:

Thanks again for the constructive comments and the time you dedicate to the paper!

**Since the discussion is about to close (Dec. 2nd)**, we would be grateful if you would kindly let us know of any other concerns and if we could further assist in clarifying any other issues.

Thanks a lot, and with sincerest best wishes

Submission 1274 Authors

---

### Meta-Review · Area_Chair_PhwQ · 2024-12-16

**Metareview:**

This paper receives final ratings of 6,6,6,8,6,3,6. The AC follows the majority of the reviewers to accept the paper. The reasons for accepting the paper are: 1) Novelty of the method and there are clear motivations in the design of each component. The proposed method is simple and efficient. A thoughtful training strategy is proposed. 2) The multi-view video generation is a novel task. 3) The proposed approach solves the data scarcity issue. 4) The experimental results show that the proposed method achieved good performance both qualitatively and quantitatively. 5) There are extensive experiments and ablation studies to support the effectiveness of the proposed techniques.

The reviewer who gave a 3 did not give any strong reasons for rejection and did not respond to the discussion and clarification of the authors. Considering the reasons to accept the paper is more compelling from other reviewers, the reviewer decided to disregard the 3 and accept the paper.

**Additional Comments On Reviewer Discussion:**

The authors managed to clarify most of the doubts from the reviewers during the discussion and rebuttal phases, which lead to several reviewers raising their scores to positive ratings.

---

### Decision · Program_Chairs · 2025-01-22

Accept (Poster)